# Compress Large Language Models via Collaboration Between Learning and Matrix Approximation

**Yuesen Liao**[1*], **Zhiwei Li**[1*], **Binrui Wu**[1*], **Zihao Cheng**[1]
**Su Zhao**[2], **Shuai Chen**[2], **Weizhong Zhang**[1,3†]
[1]Fudan University, [2]Meituan Inc.
[3]Shanghai Key Laboratory of Intelligent Information Processing
{ysliao24, zwli23, brwu23, zhcheng25}@m.fudan.edu.cn
{zhaosu04, chenshuai31}@meituan.com
weizhongzhang@fudan.edu.cn

## Abstract

Sparse and low-rank matrix composite approximation has emerged as a promising paradigm for compressing large language models (LLMs), offering a more flexible pruning structure than conventional methods based solely on sparse matrices. The significant variation in weight redundancy across layers, along with the differing rank and sparsity structures of weight matrices, makes identifying the globally optimal pruning structure extremely challenging. Existing methods often depend on uniform or manually designed heuristic rules to allocate weight sparsity across layers, subsequently compressing each matrix using matrix approximation techniques. Given the above theoretical difficulty in global compression of LLMs and the limited computational and data resources available compared to the training phase, we argue that a collaboration between learning and matrix approximation is essential for effective compression. In this paper, we propose a novel LLM compression framework based on generalized bilevel optimization that naturally formulates an effective collaborative mechanism. Specifically, the outer loop frames the weight allocation task as a probabilistic optimization problem, enabling the automatic learning of both layer-wise sparsities and matrix-wise retained ranks, while the inner loop solves the corresponding sparsity and rank-constrained model compression problem via matrix approximation. Our main technical contributions include two key innovations for efficiently solving this bilevel optimization problem. First, we introduce a truncated Gaussian prior-based probabilistic parameterization integrated with a policy gradient estimator, which avoids expensive backpropagation and stabilizes the optimization process. Second, we design an adapted QR-based matrix approximation algorithm that significantly accelerates inner loop computations. Extensive experiments on Phi-3 and the LLama-2/3 family demonstrate the effectiveness of our method. Notably, it maintains over 95% zero-shot accuracy under 50% sparsity and achieves up to 2× inference speedup.

## 1 Introduction

Model compression [8, 22, 29, 31] is a widely adopted paradigm for improving the inference efficiency of large language models (LLMs). Its core principle is to reduce model size by removing redundant parameters or approximating the model weights with low-rank matrices [37] while preserving the performance. Although promising results have been repeatedly reported in the literature,

---

[*]Equal Contribution.
[†]Corresponding Author.

39th Conference on Neural Information Processing Systems (NeurIPS 2025).

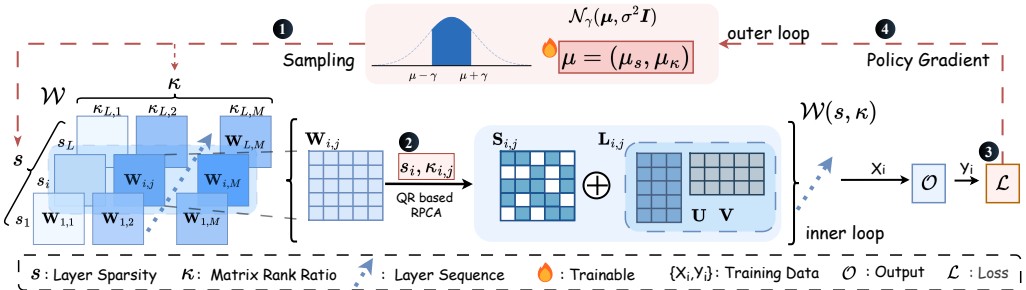

Figure 1: Process diagram of our bilevel framework. ❶: sample $s$ and $\kappa$ from $\mathcal{N}_\gamma(\boldsymbol{\mu}, \sigma^2\boldsymbol{I})$ to assign sparsity allocation for each layer; ❷: compress matrix $\mathbf{W}_{i,j} \in \mathcal{W}$ via adapted QR-based RPCA under the sparsity allocation $s_i$ and $\kappa_{i,j}$; ❸: forwardpass the compressed model $\mathcal{W}(\boldsymbol{s}, \boldsymbol{\kappa})$ and get the loss; ❹: update the distribution $\mathcal{N}_\gamma(\boldsymbol{\mu}, \sigma^2\boldsymbol{I})$ based on policy gradient estimator.

challenges remain due to the complex model architecture, vast optimization space, and limited data and computational resources compared to those available during the training stage.

In this paper, we focus on the emerging compression paradigm based on sparse and low-rank matrix composite approximation [17, 20, 44], referred to as robust principal component analysis (RPCA) [4] in the field of classical matrix analysis, which adopts a more flexible pruning structure than conventional methods based solely on sparse matrices. Existing methods [17, 44] typically adopt a uniform sparsity allocation over layers, i.e., setting an equal pruning proportion for each layer and subsequently compressing each matrix using matrix approximation techniques. Recognizing the heterogeneous redundancy across layers, recent works [18, 19, 40, 41] have introduced manually designed heuristic rules to allocate varying sparsity levels to different layers.

However, the performance of these methods is often less effective than expected. The main reason is the significant variation in weight redundancy across layers, along with differing rank and sparsity structures of weight matrices. These factors make finding the globally optimal pruning structure extremely challenging. This highlights the need for layer-wise sparsity and matrix-wise rank allocation in RPCA-based compression methods. Given the theoretical difficulty of global compression for LLMs and the limited computational resources and data compared to the training phase, we argue that collaboration between learning and matrix approximation is essential for effective compression.

In this paper, we propose a novel bilevel optimization framework [15, 30] that naturally formulates an effective collaborative mechanism. In line with recent perspectives [19, 40, 41], we adopt the view that once a global sparsity allocation is provided, the compression task can be reduced to a matrix approximation problem. Instead of metric-based heuristics [19, 41], we model the weight allocation task of **outer loop** as a probabilistic optimization problem, enabling the automatic learning of both layer-wise sparsities and matrix-wise retained ranks, while the **inner loop** solves the corresponding RPCA subproblem to obtain the sparse and low-rank decomposition under the current allocation scheme. The bilevel framework poses difficulties due to the implicit differentiation through the inner loop solutions and the substantial computational overhead of the inner RPCA problem. To address these challenges, we introduce the following two key technical innovations. First, for the outer loop, we use a truncated Gaussian prior to enable continuous probabilistic modeling within bounded support. The truncation helps stabilize training by preventing gradient explosion in low-density regions. Through this reparameterization, we apply policy gradient [32] to update the prior parameters without backpropagating through the compressed model, reducing memory overhead. Second, instead of costly SVD-based solvers [21, 49], we use an adapted QR-based matrix fitting scheme [42], which significantly accelerates inner loop computations. Our method is intuitively visualized in Figure 1. Empirical results on the Phi-3 and Llama family model show that our method consistently learns better compression configurations and achieves superior performance under various sparsities. For the LLama2-13B model, our method preserves **over 95%** MMLU accuracy under a **50% sparsity** setting, with a practical speedup of $2\times$.

Our main contributions are summarized as follows:

- We propose a bilevel optimization framework that enables effective collaboration between learning and matrix approximation for LLM compression.

- We introduce two main technical contributions: a policy gradient method based on truncated Gaussian modeling, and a QR-based RPCA algorithm for efficient matrix approximation.
- Extensive experiments demonstrate that our method consistently outperforms existing model compression methods, even under high prune rates.

## 2  Related Works

**LLM Pruning.**  LLM pruning methods can be broadly categorized into structured [12, 13, 22, 39] and unstructured [8, 9, 34, 38, 45, 46] approaches. Structured pruning removes entire components (e.g., layers, neurons, channels), with methods like LLM-Pruner [22] using gradient-based importance scores. Unstructured pruning, such as Wanda [31] and SparseGPT [8], prunes individual weights and can remove up to 30% with little accuracy drop. Wanda uses activation-aware scoring, while SparseGPT estimates Hessians for efficient weight reconstruction. However, both approaches face trade-offs between speedup and performance. Hybrid methods that combine sparsity and low-rank decomposition can better balance these aspects by integrating the strengths of both.

**Sparsity and Low Rank.**  Compression methods combining sparsity and low-rank decomposition are increasingly used for LLM compression. LoRAP [17] applies low-rank approximation to Attention matrices and enforces sparsity on MLP blocks, reflecting their distinct structures. LoSparse [20] decomposes weight matrices into low-rank factors $\mathbf{U}, \mathbf{V}$ and a sparse component $\mathbf{S}$, updating all parts while pruning $\mathbf{S}$ to meet a sparsity budget. OATS [44] and HASSLE-free [23] alternate between low-rank approximation and sparsification using fixed sparsity allocations. We focus on this class of RPCA-based compression methods as our base framework to achieve stronger performance.

**Sparsity Allocation.**  Many LLM pruning methods minimize layer-wise reconstruction loss $\|W_l X_l - \tilde{W}_l X_l\|_F^2$ and assume uniform sparsity across layers, often yielding suboptimal results. Several recent methods explore sparsity allocation strategies[26, 40]. FLAP [2] allocates sparsity based on fluctuation scores, OWL [41] leverages activation outliers, DSA [18] searches for optimal allocation functions, and ALS [19] formulates the problem as linear programming. However, these methods depend on fixed validation sets and lack joint optimization with training. In contrast, we formulate sparsity allocation as a learnable optimization problem driven by training data, while treating pruning as a matrix approximation task solvable by existing frameworks.

## 3  Preliminary

**RPCA Framework for Matrix Approximation.** As we adopt RPCA [44] as a base solver in our proposed compression method, we present its basics as follows. Given a weight matrix $\mathbf{W} \in \mathbb{R}^{m \times n}$, the target sparsity $K$ and rank $r$, RPCA approximates $\mathbf{W}$ as the sum of a low-rank matrix $\mathbf{L}$ and a sparse matrix $\mathbf{S}$ by solving the following optimization problem:

$$\min_{\mathbf{L}, \mathbf{S}} \|\mathbf{W} - \mathbf{L} - \mathbf{S}\|_F^2 \quad s.t. \quad \text{rank}(\mathbf{L}) \leq r, \ \|\mathbf{S}\|_0 \leq K. \tag{1}$$

Problem 1 is usually solved via alternating optimization, with the following update rules:

$$\begin{cases} \mathbf{L}_{t+1} = \text{TruncatedSVD}(\mathbf{W} - \mathbf{S}_t, \, r), \\ \mathbf{S}_{t+1} = \mathcal{P}_\omega(\mathbf{W} - \mathbf{L}_{t+1}), \end{cases} \tag{2}$$

where $\text{TruncatedSVD}(\mathbf{W} - \mathbf{S}, r)$ denotes the rank-$r$ approximation of matrix $\mathbf{W} - \mathbf{S}$ obtained by retaining only the top-$r$ singular values and their corresponding singular vectors. The operator $\mathcal{P}_\omega(\cdot)$ denotes the projection into the feasible set $\omega$ of the sparse matrix $\mathbf{S}$, i.e., $\omega \triangleq \{\mathbf{S} : \|\mathbf{S}\|_0 \leq K, \mathbf{S} \in \mathbb{R}^{m \times n}\}$. Typically, this projection enforces a sparsity constraint by retaining only the top-$K$ largest-magnitude entries and setting the rest to zero. Following Wanda and OATS, we apply a diagonal scaling matrix $\mathbf{D} = \sqrt{\text{diag}(\boldsymbol{X}^\top \boldsymbol{X})} \in \mathbb{R}^{n \times n}$ to the weight matrix and perform approximation on $\mathbf{WD}$, where $\boldsymbol{X}$ denotes the input activation. This is used by default unless otherwise specified.

**Discussion.** It is important to note that the above solver is not efficient enough due to the expensive SVD process and cannot be directly adapted to develop our compression method. Therefore, in this paper, we introduce an efficient RPCA algorithm based on QR decomposition, detailed in Section 4.3.

# 4 Method

In this section, we present the details of our method. Section 4.1 introduces the overall design of the bilevel optimization framework. Section 4.2 describes our first technical contribution: we propose a truncated Gaussian prior and integrate it with policy gradient estimator to stabilize the training process and avoid expensive implict differentiation. Section 4.3 presents our second technical contribution: an efficient RPCA algorithm adapted from QR decomposition.

## 4.1 Bilevel Framework

We introduce our bilevel optimization framework for LLM compression, which formulates an effective collaborative mechanism between learning and matrix approximation. The inner loop performs model compression by solving an RPCA problem under a specific sparsity allocation scheme given by the outer loop. The outer loop formulates the learning problem of sparsity allocation into a probabilistic optimization task, enabling the automatic learning of both layer-wise sparsities and matrix-wise retained ranks based on the model compressed by the inner loop. The workflow is shown in Figure 1.

**Inner Loop.** We first describe how a given allocation scheme determines the sparsity structure of each matrix in the model. To capture differences in parameter redundancy across layers, we allocate a sparsity ratio $s_i$ to each layer, indicating the proportion of parameters to be pruned. For all matrices $\{\mathbf{W}_{i,j}\}_{j=1}^M$ in layer $i$ that lie in $\mathbb{R}^{m \times n}$ , the total number of retained parameters after compression is $mn(1 - s_i)$. We let $\kappa_{i,j}$ be the proportion of parameters allocated to $\mathbf{W}_{i,j}$ for the low rank matrix $\mathbf{L}_{i,j}$, i.e., we assign $mn(1 - s_i)\kappa_{i,j}$ parameters to $\mathbf{L}_{i,j}$. This yields the target rank and sparsity:

$$r_{i,j} = \frac{mn(1 - s_i)\kappa_{i,j}}{m + n}, \quad K_{i,j} = mn(1 - s_i)(1 - \kappa_{i,j}). \tag{3}$$

We group all $s_i$ and $\kappa_{i,j}$ into two vectors $(\boldsymbol{s}, \boldsymbol{\kappa})$ and compress each matrix accordingly by solving a series of RPCA problems described in Section 3. That is, we can obtain a set of RPCA problems presented in the definition below.

**Definition 1.** *The* RPCA *decomposition of the full parameter set* $\mathcal{W} \triangleq \{\mathbf{W}_{i,j} | i \in [1, L], j \in [1, M]\}$ *under sparsity allocation scheme* $(\boldsymbol{s}, \boldsymbol{\kappa})$ *is denoted as*

$$\mathrm{RPCA}(\mathcal{W}, \boldsymbol{s}, \boldsymbol{\kappa}) = \left\{ \tilde{\mathbf{W}}_{i,j} = \underset{\substack{\|\mathbf{S}_{i,j}\|_0 \leq K_{i,j} \\ \mathrm{rank}(\mathbf{L}_{i,j}) \leq r_{i,j}}}{\arg\min} \|\mathbf{W}_{i,j} - \mathbf{L}_{i,j} - \mathbf{S}_{i,j}\|_F^2 \ \Big| \ \mathbf{W}_{i,j} \in \mathcal{W} \right\},$$

*where each matrix* $\mathbf{W}_{i,j}$ *is decomposed into a low-rank component* $\mathbf{L}_{i,j}$ *and a sparse component* $\mathbf{S}_{i,j}$ *with target rank* $r_{i,j}$ *and sparsity budget* $K_{i,j}$ *computed from Eqn. (3).*

**Outer Loop.** In the outer loop, we begin by modeling the allocation scheme $(\boldsymbol{s}, \boldsymbol{\kappa})$ using a suitable probabilistic distribution $p(\cdot|\boldsymbol{\theta})$ parameterized by $\boldsymbol{\theta}$. A sparsity allocation is sampled from this distribution and passed into the inner loop to generate a compressed model $\mathcal{W}(\boldsymbol{s}, \boldsymbol{\kappa})$. The performance of the resulting model is then evaluated using a loss function. The overall objective is to minimize the expected loss over sampled allocation schemes. To this end, we optimize the probability parameters $\boldsymbol{\theta}$ via gradient-based methods, enabling the framework to adaptively explore and refine sparsity patterns that lead to improved model performance.

Therefore, the overall bilevel optimization framework can be formulated as follows:

$$\min_{\boldsymbol{\theta} \in \mathcal{C}} \quad \mathbb{E}_{(\boldsymbol{s}, \boldsymbol{\kappa}) \sim p(\cdot|\boldsymbol{\theta})} \mathcal{L}(\mathcal{W}(\boldsymbol{s}, \boldsymbol{\kappa})) = \frac{1}{N} \sum_{i=1}^N \ell(f(\boldsymbol{x}_i, \mathcal{W}(\boldsymbol{s}, \boldsymbol{\kappa})), \boldsymbol{y}_i),$$

$$s.t. \quad \mathcal{W}(\boldsymbol{s}, \boldsymbol{\kappa}) = \mathrm{RPCA}(\mathcal{W}, \boldsymbol{s}, \boldsymbol{\kappa}), \tag{4}$$

where $\mathcal{C}$ is the feasible region for $\boldsymbol{\theta}$ to control the sparsity, which will be specified in Section 4.2. $\{(\boldsymbol{x}_i, \boldsymbol{y}_i)\}_{i=1}^N$ represents the training dataset, $f(\cdot, \mathcal{W}(\boldsymbol{s}, \boldsymbol{\kappa}))$ is the compressed model under allocation scheme $(\boldsymbol{s}, \boldsymbol{\kappa})$, and $\ell(\cdot, \cdot)$ denotes the loss function.

**Challenges.** This bilevel optimization framework presents two main challenges. First, the outer objective is hard to optimize due to the implicit differentiation through the inner loop, requiring appropriate gradient estimators and a well-designed probabilistic model $p(\cdot \mid \boldsymbol{\theta})$. Second, repeatedly solving RPCA problems in the inner loop is computationally expensive. In the following sections, we introduce our proposed techniques to address these challenges.

## 4.2 Outer Optimization

**Policy Gradient.** To address the difficulty of computing gradients with respect to the parameter $\boldsymbol{\theta}$, we adopt a policy gradient estimator. This approach avoids implicit differentiation by directly computing gradients based on the loss function. The derivation is as follows:

$$\nabla_{\boldsymbol{\theta}} \mathbb{E}_{p(\boldsymbol{s},\boldsymbol{\kappa}|\boldsymbol{\theta})} \left[ \mathcal{L}(\mathcal{W}(\boldsymbol{s},\boldsymbol{\kappa})) \right] = \mathbb{E}_{p(\boldsymbol{s},\boldsymbol{\kappa}|\boldsymbol{\theta})} \left[ \mathcal{L}(\mathcal{W}(\boldsymbol{s},\boldsymbol{\kappa})) \cdot \nabla_{\boldsymbol{\theta}} \log p(\boldsymbol{s},\boldsymbol{\kappa}|\boldsymbol{\theta}) \right]. \tag{5}$$

In practice, we sample a mini-batch $\mathcal{B}$, evaluate the loss of the compressed model under each sampled allocation $(\boldsymbol{s},\boldsymbol{\kappa})$, and compute the policy gradient of the parameters $\boldsymbol{\theta}$ as: $g_{\boldsymbol{\theta}} = \mathcal{L}_{\mathcal{B}}(\mathcal{W}(\boldsymbol{s},\boldsymbol{\kappa})) \cdot \nabla_{\boldsymbol{\theta}} \log p(\boldsymbol{s},\boldsymbol{\kappa}|\boldsymbol{\theta})$. This yields an unbiased estimator; the proof is provided in Appendix D.1.

**Remark 1.** *Policy gradient methods are known to suffer from high variance due to the stochastic nature of the sampling process, which can lead to instability during training. To mitigate this issue, we subtract a control variate $\mathcal{L}_{\mathcal{B}}(\mathcal{W}(\boldsymbol{s}',\boldsymbol{\kappa}')) \cdot \nabla_{\boldsymbol{\theta}} \log p(\boldsymbol{s},\boldsymbol{\kappa}|\boldsymbol{\theta})$, which has zero mean but is highly correlated with the original gradient. Here, $(\boldsymbol{s}',\boldsymbol{\kappa}')$ is an independent sample drawn from the same distribution as $(\boldsymbol{s},\boldsymbol{\kappa})$. This variance-reduction technique leads to the final gradient estimator:*

$$\boldsymbol{g}_{\boldsymbol{\theta}}^{vr} = \left[ \mathcal{L}_{\mathcal{B}}(\mathcal{W}(\boldsymbol{s},\boldsymbol{\kappa})) - \mathcal{L}_{\mathcal{B}}(\mathcal{W}(\boldsymbol{s}',\boldsymbol{\kappa}')) \right] \cdot \nabla_{\boldsymbol{\theta}} \log p(\boldsymbol{s},\boldsymbol{\kappa}|\boldsymbol{\theta}). \tag{6}$$

**Truncated Gaussian Distribution.** Computing policy gradients requires $\nabla_{\boldsymbol{\theta}} \log p(\boldsymbol{s},\boldsymbol{\kappa}|\boldsymbol{\theta})$. Gaussian distributions are often used for convenience, but their support is unbounded, conflicting with the constraints (e.g., sparsity ratio in $[0,1]$). Moreover, shrinking variance for convergence can cause gradient explosion. To address these issues, we employ a truncated Gaussian distribution $\mathcal{N}_{\gamma}(\mu,\sigma^2)$ for probabilistic modeling, which restricts the Gaussian distribution $\mathcal{N}(\mu,\sigma^2)$ to the interval $[\mu-\gamma, \mu+\gamma]$. This truncation limits the sampling range and provides bounded support, thereby stabilizing training and facilitating policy gradient computation. For a random variable $x \sim \mathcal{N}_{\gamma}(\mu,\sigma^2)$, its probability density function (PDF) is given by:

$$p(x;\mu,\sigma^2,\gamma) = \begin{cases} \dfrac{1}{\sigma} \cdot \dfrac{\phi\left(\frac{x-\mu}{\sigma}\right)}{\Phi\left(\frac{\gamma}{\sigma}\right) - \Phi\left(\frac{-\gamma}{\sigma}\right)}, & \text{for } x \in [\mu-\gamma, \, \mu+\gamma], \\ 0, & \text{otherwise,} \end{cases}$$

where $\phi$ and $\Phi$ denote the PDF and CDF of the standard Gaussian distribution $\mathcal{N}(0,1)$, respectively. The detailed sampling method for the truncated Gaussian is described in Appendix C.2.

**Remark 2.** *For the $p(x;\mu,\sigma^2,\gamma)$, its parameter vector is $[\mu,\sigma^2,\gamma]$. To avoid the gradient explosion during training, we fix the variance $\sigma^2$. In addtion, to control the range of $x$ and ensure convergence, we manually reduce $\gamma$ according to the annealing schedule [50]. More details are provided in Appendix A.2. Therefore, the only trainable parameter of $p(\cdot)$ is $\mu$, i.e., $\boldsymbol{\theta} = \mu$. For simplicity, we do not distinguish between $\mu$ and $\boldsymbol{\theta}$ in the remainder of this section.*

Each element in gradient $\nabla_{\boldsymbol{\mu}} \log p(\boldsymbol{s},\boldsymbol{\kappa}|\boldsymbol{\mu},\sigma^2,\gamma)$ can be computed using the lemma below.

**Lemma 1.** *Let a random variable $x$ follow the truncated Gaussian distribution $x \sim \mathcal{N}_{\gamma}(\mu,\sigma^2)$. The gradient of the log-density with respect to the mean parameter $\mu$ is given by:*

$$\nabla_{\mu} \log p(x|\mu,\sigma^2,\gamma) = \nabla_{\mu} \log \left( \frac{1}{\sigma} \cdot \frac{\phi\left(\frac{x-\mu}{\sigma}\right)}{\Phi\left(\frac{\gamma}{\sigma}\right) - \Phi\left(\frac{-\gamma}{\sigma}\right)} \right) = \nabla_{\mu} \log \left( \phi\left(\frac{x-\mu}{\sigma}\right) \right) = \frac{x-\mu}{\sigma^2}. \tag{7}$$

Combining Eqn. (6) and Eqn. (7), we compute the gradient $\boldsymbol{g}_{\boldsymbol{\mu}}^{vr}$ with $\boldsymbol{\mu} = (\boldsymbol{\mu_s}, \boldsymbol{\mu_\kappa})$. Recall that $\boldsymbol{\mu}$ is the mean of $(\boldsymbol{s},\boldsymbol{\kappa})$, its feasible region can be defined as $\mathcal{C} \triangleq \left\{ \boldsymbol{\mu} : \|\boldsymbol{\mu_s}\|_1 \geq \rho L, \ \boldsymbol{\mu} \in [0,1]^{L+LM} \right\}$. Since $\mathcal{C}$ can be rewriten as $\mathcal{C} = \left\{ \boldsymbol{\mu} : \mathbf{1}^\top \boldsymbol{\mu_s} \geq \rho L, \ \boldsymbol{\mu} \in [0,1]^{L+LM} \right\}$, which is convex, we can project $\boldsymbol{\mu}$ onto $\mathcal{C}$ using projection $\mathrm{proj}_{\mathcal{C}}(\cdot)$ after gradient descent. See Appendix C.1 for details.

---

**Algorithm 1** Bilevel Optimization Framework

---

**Input:** Model weights $\mathcal{W}$, over all prune rate $\rho$, rank ratio $\kappa_0$, parameter $\sigma^2$ and $\gamma$, learning rate $\eta$
1: Initialize parameters $\boldsymbol{\mu} = (\boldsymbol{\mu_s}, \boldsymbol{\mu_\kappa})^3$
2: **for** each iteration **do**
3:      Sample a mini batch $\mathcal{B}$
4:      Reduce $\gamma$ according to annealing schedule
5:      Sample $(\boldsymbol{s}^{(i)}, \boldsymbol{\kappa}^{(i)})$ from $p(\boldsymbol{s}, \boldsymbol{\kappa}|\boldsymbol{\mu}, \sigma^2, \gamma), i = 1, 2$
6:      Apply RPCA, i.e., Algorithm 2, to obtain the compressed weights $\mathcal{W}(\boldsymbol{s}^{(i)}, \boldsymbol{\kappa}^{(i)}), i = 1, 2$
7:      Compute $\mathcal{L}_\mathcal{B}(W(\boldsymbol{s}^{(i)}, \boldsymbol{\kappa}^{(i)})), i = 1, 2$, and the gradient:

$$\boldsymbol{g}_{\boldsymbol{\mu}}^{vr} = \left[\mathcal{L}_\mathcal{B}(\mathcal{W}(\boldsymbol{s}^{(1)}, \boldsymbol{\kappa}^{(1)})) - \mathcal{L}_\mathcal{B}(\mathcal{W}(\boldsymbol{s}^{(2)}, \boldsymbol{\kappa}^{(2)}))\right] \cdot \nabla_{\boldsymbol{\mu}} \log p(\boldsymbol{s}^{(1)}, \boldsymbol{\kappa}^{(1)}|\boldsymbol{\mu}, \sigma^2, \gamma)$$

8:      Update: $\boldsymbol{\mu} \leftarrow \text{proj}_\mathcal{C}(\boldsymbol{\mu} - \eta \boldsymbol{g}_{\boldsymbol{\mu}}^{vr})$
9: **end for**
**Output:** Compressed weights $\mathcal{W}(\boldsymbol{\mu_s}, \boldsymbol{\mu_\kappa})$

---

### 4.3 Inner Optimization

**QR-based RPCA algorithm.** Conventional RPCA algorithms repeatedly perform costly SVD computations, resulting in high computational overhead. In the inner loop, we follow and adapt the method proposed in [42], replacing SVD with a more efficient QR-based algorithm. Specifically, noting that the low-rank matrix $\mathbf{L}$ in Problem 1 can be factorized as the product of two matrices $\mathbf{U} \in \mathbb{R}^{m \times r}$ and $\mathbf{V} \in \mathbb{R}^{r \times n}$, where $r$ is the target rank, we obtain the following reformulation:

$$\min_{\mathbf{U}, \mathbf{V}, \mathbf{S}} \|\mathbf{W} - \mathbf{U}\mathbf{V} - \mathbf{S}\|_F^2 \quad s.t. \quad \text{rank}(\mathbf{U}\mathbf{V}) \leq r, \|\mathbf{S}\|_0 \leq K \tag{8}$$

This problem can be solved by alternating minimization over $\mathbf{U}$, $\mathbf{V}$, and $\mathbf{S}$, yielding the update rules:

$$\begin{cases} \mathbf{U}_{t+1} = (\mathbf{W} - \mathbf{S}_t)\mathbf{V}_t^\top (\mathbf{V}_t \mathbf{V}_t^T)^\dagger, \\ \mathbf{V}_{t+1} = (\mathbf{U}_{t+1}^\top \mathbf{U}_{t+1})^\dagger \mathbf{U}_{t+1}^\top (\mathbf{W} - \mathbf{S}_t), \\ \mathbf{S}_{t+1} = \mathcal{P}_\omega(\mathbf{W} - \mathbf{U}_{t+1}\mathbf{V}_{t+1}). \end{cases} \tag{9}$$

The optimization objective only depends on the product $\mathbf{U}\mathbf{V}$, rather than the specific factorization. Therefore, we aim to find any pair $(\mathbf{U}', \mathbf{V}')$ such that $\mathbf{U}'\mathbf{V}' = \mathbf{U}\mathbf{V}$. This insight allows us to reinterpret the optimization as a projection problem.

$$\mathbf{U}_{t+1}\mathbf{V}_{t+1} = \mathbf{U}_{t+1}\left(\mathbf{U}_{t+1}^\top \mathbf{U}_{t+1}\right)^\dagger \mathbf{U}_{t+1}^\top (\mathbf{W} - \mathbf{S}_t) = \Pi_{\mathcal{C}(\mathbf{U}_{t+1})}(\mathbf{W} - \mathbf{S}_t), \tag{10}$$

where $\Pi_{\mathcal{C}(\mathbf{U}_{t+1})}$ denotes the orthogonal projection onto the column space of $\mathbf{U}_{t+1}$.

Since $\left(\mathbf{V}_t \mathbf{V}_t^\top\right)^\dagger$ is full-rank, the column space of $\mathbf{U}_{t+1}$ is equivalent to that of $(\mathbf{W} - \mathbf{S}_t)\mathbf{V}_t^\top$. Thus, we perform a QR decomposition on this matrix:

$$(\mathbf{W} - \mathbf{S}_t)\mathbf{V}_t^\top = \mathbf{Q}_t \mathbf{R}_t, \tag{11}$$

where $\mathbf{Q}_t \in \mathbb{R}^{m \times r}$ is orthonormal and spans the column space of $\mathbf{U}_{t+1}$, leading to the expression:

$$\mathbf{U}_{t+1}\mathbf{V}_{t+1} = \mathbf{Q}_t \mathbf{Q}_t^\top (\mathbf{W} - \mathbf{S}_t), \tag{12}$$

and we accordingly set the update rules as:

$$\mathbf{U}_{t+1} := \mathbf{Q}_t, \quad \mathbf{V}_{t+1} := \mathbf{Q}_t^\top (\mathbf{W} - \mathbf{S}_t). \tag{13}$$

We present the RPCA algorithm based on QR decomposition in Appendix C.3 Algorithm 2.

**Remark 3.** *The QR-based method reduces the per-iteration complexity to $\mathcal{O}(mr^2)$, much lower than the $\mathcal{O}(mn\min(m, n))$ cost of SVD when $r \ll \min(m, n)$, while maintaining approximation quality.*

**Remark 4.** *Rather than reinitializing the inner RPCA subroutine from scratch at each iteration, we warm-start the optimization using the previous solution and adjust the low-rank factors $\mathbf{U}, \mathbf{V}$ incrementally according to the rank update $\Delta r$, thereby improving efficiency and stability.*

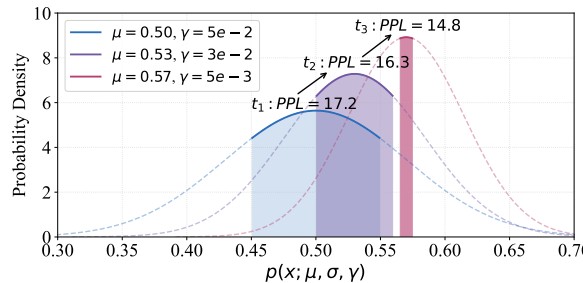
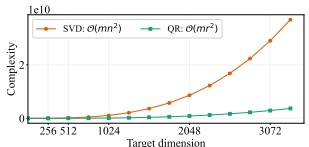

(a) Trajectory of the 25-th layer's sparsity during optimization.

(b) Complexity of QR and SVD.

| Decomposition Type | Time Cost (80 iters) |
|---|---|
| SVD | 5 h |
| QR | 4 min |

(c) Time cost of QR and SVD.

Figure 2: (a) the variation of the truncated Gaussian distribution as training progresses reflects how the structure gradually learns to approach the optimum. (b) due to $r \ll \min(m, n)$, QR has a lower complexity. (c) QR decomposition provides significant acceleration in practical implementation.

By integrating the truncated Gaussion prior with policy gradient estimator and the QR-based RPCA solver, we obtain the complete bilevel optimization framework presented in Algorithm 1. For the last step, since the range of $(s, \kappa)$ is vanished, we compress the model under $(\mu_s, \mu_\kappa)$ without sampling. We present the convergence analysis of the bilevel optimization framework in Appendix D.4.

## 5 Experiments

In Section 5.1, we introduce the overall experimental setup and the baselines used for comparison. Section 5.2 presents the pruning performance across multiple LLMs. In Section 5.3, we explores the effectiveness of our method under high sparsity settings. Finally, Section 5.4 conducts ablation studies to validate the effectiveness of each component in our framework.

### 5.1 Experimental Setups

**Models.** For our main experiments, we select representative models from two prominent open-source architectures: the Phi family (specifically Phi-3-mini [1]) and the Llama family (including Llama2-7B, Llama2-13B [33], and Llama3-8B [7]).

**Baseline.** We select SOTA compression methods as baselines, including unstructured pruning methods such as SparseGPT [8] and Wanda [31], as well as RPCA based approaches like OATS and QR, where QR refers to using only the inner QR-based algorithm in Section 4.3, without the bilevel optimization framework.[44]. Our experiments compare these compression methods at low prune rates ($\leq 50\%$) and validate the effectiveness of sparsity allocation at higher prune rates ($\geq 60\%$).

**Configurations.** We use C4 [27] as the training dataset, with batch size set to 32, and length set to 256. In addition, the inner-level optimization employs the fixed 32 samples as the calibration dataset. Gamma is set from 0.05 to 0.005. In training, we use the Adam optimizer [16] and set the learning rate to 1e-2. The experiments are all completed with one single 80GB NVIDIA A100.

**Evaluation.** We use LM-evaluation-harness [10] to evaluate the performance after pruning. The main benchmarks include: 1) WikiText2 [24] perplexity, 2) zero-shot tasks (including PIQA [3], HellaSwag [43], Winogrande [28], OpenBookQA [25], RTE [35], BoolQ [5], ARC-e and ARC-c [6]), and 3) few shot tasks, like MMLU [14]. In addition, we test the CPU inference speedup of the pruned model on Intel(R) Xeon(R) Platinum 8369B CPU @ 2. 90GHz with 32 cpu cores.

### 5.2 Comparison of Compression Methods

Table 1 presents the main results, comparing the performance of different models using various compression methods across multiple prune rates. Our approach achieves top performance across all three task types. Notably, on the WikiText2 benchmark, it reduces Phi-3-mini's perplexity by about 5% over the SOTA OATS at 50% sparsity. Results on MMLU and zero-shot accuracy further show that RPCA-based methods (OATS, QR, and ours) outperform methods relying solely on sparse matrix

---

[3]The initialization details of $\mu_s$ and $\mu_\kappa$ are provided in Appendix A.2.

Table 1: Performance comparison across various methods and models with different prune rates. The best performance for each prune rate is in **bold**.

| Prune rate | Method | Phi-3-mini | | | Llama2-7B | | | Llama2-13B | | | Llama3-8B | | |
|---|---|---|---|---|---|---|---|---|---|---|---|---|---|
| | | ↓WikiText2 | ↑MMLU | ↑zero-shot | WikiText2 | MMLU | zero-shot | WikiText2 | MMLU | zero-shot | WikiText2 | MMLU | zero-shot |
| 0% | Dense | 9.50 | 70.34 | 71.99 | 8.79 | 50.12 | 66.27 | 7.91 | 56.41 | 68.72 | 10.18 | 64.97 | 69.71 |
| 30% | SparseGPT | 11.19 | 68.31 | 70.36 | 9.29 | 49.10 | 64.99 | 8.29 | 54.48 | 68.35 | 9.71 | 64.25 | 69.08 |
| | Wanda | 10.71 | 67.63 | 70.66 | 9.23 | 47.56 | 65.31 | 8.29 | 55.1 | 68.23 | 9.71 | 63.67 | 68.63 |
| | OATS | 10.27 | 68.84 | 71.48 | 9.06 | 49.98 | 65.11 | 8.11 | 55.97 | **68.76** | 9.59 | 65.22 | 69.34 |
| | QR | 10.34 | 68.35 | 71.06 | 9.10 | 50.02 | 65.89 | 8.17 | 53.99 | 68.36 | **8.00** | 63.28 | **70.00** |
| | Ours | **9.98** | **69.60** | **71.51** | **8.83** | 50.10 | 66.19 | **8.02** | **56.13** | 68.74 | 8.06 | **65.37** | 69.82 |
| 40% | SparseGPT | 13.03 | 63.47 | 69.18 | 9.94 | 45.52 | 64.13 | 8.85 | 54.48 | 68.35 | 10.01 | 60.91 | 67.58 |
| | Wanda | 12.59 | 64.15 | 68.80 | 9.86 | 44.8 | 64.70 | 8.77 | 53.65 | 68.06 | 9.74 | 60.33 | 67.04 |
| | OATS | 11.53 | 65.75 | 70.04 | 9.53 | 47.21 | 65.63 | 8.45 | 55.25 | 68.16 | 9.24 | 62.46 | **68.68** |
| | QR | 11.67 | 64.28 | 69.98 | 9.56 | 46.64 | 64.53 | 8.56 | 54.96 | 68.6 | 8.70 | 61.94 | 68.29 |
| | Ours | **11.03** | **65.92** | **70.56** | **9.14** | **47.62** | 65.86 | **8.31** | **55.70** | 68.64 | 8.59 | 62.53 | 68.62 |
| 50% | SparseGPT | 16.80 | 53.22 | 66.36 | 11.66 | 41.94 | 62.69 | 10.21 | 48.91 | 66.68 | 11.95 | 53.60 | 64.66 |
| | Wanda | 17.23 | 54.57 | 65.03 | 11.43 | 37.16 | 62.53 | 10.05 | 49.59 | 66.28 | 12.36 | 49.83 | 63.27 |
| | OATS | 15.18 | 59.99 | 68.41 | 10.87 | 44.7 | 63.49 | 9.49 | 52.44 | 67.77 | 10.87 | 56.46 | 65.71 |
| | QR | 15.30 | 58.28 | 67.48 | 10.86 | 42.53 | 63.09 | 9.58 | 53.31 | 67.65 | 10.70 | 55.30 | 65.54 |
| | Ours | **14.87** | **60.57** | **69.37** | **10.49** | **46.10** | **64.08** | **9.23** | **53.79** | **67.92** | **10.18** | **56.97** | **66.28** |

compression. Our method further excels by adaptively allocating sparsity. Interestingly, Llama3-8B even surpasses its unpruned version at low prune rates, suggesting that pruning redundant weights may be beneficial. Moreover, the QR-based method with uniform allocation achieves performance close to OATS while requiring only 1/20 of its runtime, highlighting both efficiency and effectiveness, thereby offering a promising direction for scalable LLM compression.

## 5.3 Comparison with OWL Sparsity Allocation

We further investigate the effectiveness of different sparsity allocation strategies under high prune rates. To ensure a fair comparison, we adopt the QR decomposition introduced in Section 4.3 as the sole compression algorithm. Table 2 compares the results of various sparsity allocation strategies, where uniform denotes applying the same prune rates to all layers, and OWL leverages outlier information to allocate sparsity. It can be observed that our method leads to better performance, providing valuable insights for future research on high sparsity compression.

Table 2: Comparison of sparsity allocation methods under high sparsity.

| Prune rate | Method | Phi-3-mini | | | Llama2-7B | | | Llama3-8B | | |
|---|---|---|---|---|---|---|---|---|---|---|
| | | ↓PPL | ↑MMLU | ↑zero-shot | PPL | MMLU | zero-shot | PPL | MMLU | zero-shot |
| 60% | Uniform | 48.8 | 38.22 | 56.01 | 16.92 | 32.99 | 58.61 | 20.03 | 34.06 | 56.62 |
| | OWL | 35.37 | 51.43 | 58.18 | 15.38 | 39.33 | 59.79 | 17.39 | 44.07 | 59.27 |
| | Ours | **32.46** | **52.27** | **59.24** | **15.07** | **39.78** | **60.52** | **17.03** | 43.91 | **59.34** |
| 70% | Uniform | 1375.75 | 25.5 | 40.88 | 122.9 | 24.53 | 42.49 | 111.37 | 26.8 | 41.14 |
| | OWL | 462.67 | 28.5 | 46.33 | 56.38 | 26.66 | 48.05 | 67.72 | 26.9 | 45.81 |
| | Ours | **208.7** | **30.27** | **49.53** | **47.21** | **29.38** | **51.62** | **58.34** | **27.6** | **47.04** |

## 5.4 Ablation Study

We conduct ablation studies on the components of the proposed bilevel optimization framework using Phi-3-mini with 50% sparsity. We first compare the effects of layer-wise sparsity and matrix rank ratio allocation on the performance of the compressed model, where w/o Rank denotes only sparsity allocation without rank ratio allocation, w/o Sparsity denotes only rank ratio allocation without sparsity allocation, and w/o Rank & Sparsity denotes neither allocation being applied. The results, shown in Table 3, demonstrate that rank ratio allocation has a more significant impact on performance, a factor that has been overlooked in other sparsity allocation methods. We also compare the runtime of different RPCA solvers in Table 2c, where QR-based method is significantly faster than SVD. Additionally, we further compare the performance and runtime of the two decomposition methods under varying iteration counts, as shown in Figure 3. At 80 iterations, both methods achieve similar zero-shot accuracy, while the QR-based method completes in just a few minutes.

## 6 Further Analysis

In this section, we discuss the allocation results and actual acceleration effects of various types of model compression, as well as the performance of our bilevel optimization framework when combined with other metric-based compression methods.

Table 3: Ablation study on layer-wise sparsity and matrix rank ratio allocation.

| Alloc type | ↓ Perplexity | ↑ MMLU | ↑ zero-shot |
|---|---|---|---|
| Sparsity & Rank | 14.87 | 60.57 | 69.37 |
| w/o Rank | 14.94 | 59.94 | 68.48 |
| w/o Sparsity | 14.89 | 60.04 | 68.93 |
| w/o Rank & Sparsity | 15.30 | 58.28 | 67.48 |

Figure 3: Impact of the number of iterations on accuracy and time cost.

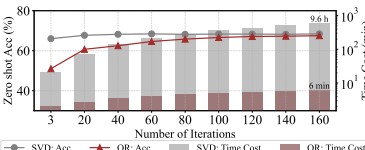

## 6.1 Allocation Visualization

Figure 4 illustrates the allocation results of Phi-3-mini obtained by our bilevel framework. The prune rate gradually increases across layers. This reflects the variation in parameter redundancy across layers. Regarding rank ratios across matrices, our method accurately captures the heterogeneous low-rankness. For matrices in MLP, fewer budgets are allocated to the low-rank component, allowing more flexibility for the sparse component to preserve model fitting capacity. Conversely, for attention blocks, our method allocates more rank budget to fully exploit the underlying structure.

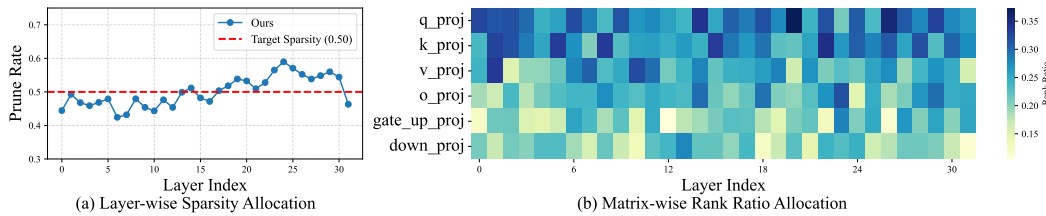

(a) Layer-wise Sparsity Allocation

(b) Matrix-wise Rank Ratio Allocation

Figure 4: Left: Layer-wise sparsity allocation $s$; Right: Matrix-wise rank ratio allocation $\kappa$.

## 6.2 CPU SpeedUp

Following the settings in OATS and OWL, we use the DeepSparse engine to evaluate the actual CPU acceleration achieved by various compression methods on the Llama2-13B model. As shown in Table 4, RPCA-based methods yield greater speedups (up to $1.99\times$ at 50% sparsity) thanks to structured low-rank components, and also outperform purely sparse methods in accuracy.

Table 4: Throughput and speedup comparison among different prune types.

| Prune rate | Prune type | Method | ↑ Throughput ($\mathcal{B}$/s) | ↑ Speedup ($\times$) | ↓ Perplexity |
|---|---|---|---|---|---|
| 0% | Dense | – | 2.38 | 1.00 | 7.91 |
| 40% | Unstructured | Wanda | 2.93 | 1.23 | 8.77 |
| | Low rank & Sparsity | Ours | **3.92** | **1.64** | **8.31** |
| 50% | Unstructured | Wanda | 3.99 | 1.68 | 10.05 |
| | Semi-unstructured (2:4) | Wanda | 4.38 | 1.84 | 16.53 |
| | Low rank & Sparsity | Ours | **4.75** | **1.99** | **9.23** |

## 6.3 Integration with Other Compression Methods

To extend our framework beyond low-rank and sparse decomposition, we apply it to metric-based pruning methods like Wanda and SparseGPT. Table 5 shows results on Phi-3-mini at 50% sparsity. Compared to uniform and OWL-based allocations, our method consistently outperforms both. This demonstrates our approach's potential as a general sparsity allocation mechanism across compression methods, providing new insights for model compression. The performance improvement brought by our method is less pronounced compared to the RPCA-based approach, suggesting that the latter presents a more challenging problem that requires accurate sparsity allocation optimization.

Table 5: Integration with other compression methods.

| Base method | Alloc method | ↓ Perplexity | ↑ MMLU | ↑ zero-shot |
|---|---|---|---|---|
| Wanda | Uniform | 17.23 | 54.57 | 65.03 |
| | OWL | 16.22 | 55.27 | 65.97 |
| | Ours | **16.01** | **55.78** | **66.14** |
| SparseGPT | Uniform | 16.80 | 53.22 | 66.36 |
| | OWL | 17.39 | 56.35 | 65.95 |
| | Ours | **16.18** | **56.86** | **66.63** |

## 7 Conclusion

In this work, we propose a bilevel optimization framework that unifies learning and matrix approximation for LLM compression. By formulating sparsity and rank allocation as a probabilistic optimization problem and solving the matrix approximation subtask via RPCA, our method effectively captures weight redundancy structures. We introduce a truncated Gaussian prior for probabilistic parameterization, combined with a policy gradient estimator, which avoids implicit differentiation through the inner loop and stabilizes training. Additionally, we design a QR-based RPCA solver that significantly accelerates the inner loop computation. Our collaborative mechanism offers a new perspective and practical methodology for advancing efficient and effective model compression.

## 8 Acknowledgements

This work was supported by the National Nature Science Foundation of China (62472097), Shanghai Municipal Science and Technology Commission (Grant No.24511106102), Fudan Kunpeng&Ascend Center of Cultivation and AI for Science Foundation of Fudan University (FudanX24AI028). The computations in this research were performed on the CFFF platform of Fudan University.

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
