# OpenReview forum: "Compress Large Language Models via  Collaboration Between Learning and Matrix Approximation"
_NeurIPS.cc/2025/Conference — NeurIPS 2025 poster_

### Official Review · Reviewer_nSyU · 2025-06-25

**Clarity:** 2
**Significance:** 3
**Originality:** 3
**Rating:** 3
**Confidence:** 4

**Summary:**

This paper studies the problem of assigning rank and sparsity values to different layers under the sparse + low rank compression setup. They present a two step optimization problem, where the outer step decides the allocation and the inner step does the compression. A policy gradient approximation is used for the outer step, while a QR-based solver is used for the inner loop.

**Questions:**

See my comments above.

**Ethical Concerns:**

["NO or VERY MINOR ethics concerns only"]

**Final Justification:**

Though I appreciate the authors' efforts to address my concerns, as can be seen from my discussions with them, there are a lot of details that need to be hashed out, new baselines need to be added, etc. Therefore, I think the paper needs to go through a major revision and these issues cannot be addressed via a short rebuttal. Hence, I recommend the paper to be rejected.

**Limitations:**

yes

**Quality:**

2

**Strengths And Weaknesses:**

The problem that is studied is interesting from a methodological perspective, though I have doubts how useful it'll be in practice. The authors provide CPU inference results but in reality it is rare to use CPUs for LLM inference. Unless some sort of GPU acceleration can be shown, this work's application will be limited.

Also, the policy gradient part is not properly motivated, and it looks out of place (in addition to the truncated normal prior). I think the authors should provide more context and justification as to why the stochastic approximation is good in practice.

The experiments have a basic flaw. This paper, following Wanda and OATS, uses a diagonal approximation to $X^TX$, which is known to be a bad approximation, even though it simplifies the algebra. Therefore, even though the non-uniform allocation of this paper shows improvements over OATS, I'm not convinced the models from the proposed method are actually good. I would like to ask authors to provide comparisons with other sparse + low rank methods, such as https://openreview.net/pdf?id=hyN75SAJTI.

---

> ### Author Rebuttal · Authors · 2025-07-31
>
> ## W1: GPU acceleration.
> Thanks for your suggestion. In our initial experiments, we followed the evaluation protocols of OATS and OWL, which focus on CPU speed up, and therefore we did not include GPU results.
> To address this, we followed the ATP[r1] protocol and conducted GPU inference tests using `nm-vllm`. The results are presented in **Table R1**.
> Compared with unstructured methods, our approach, Low rank & Sparse, leads to more practically meaningful GPU inference acceleration.
> #### Table R1 GPU inference speedup LLaMA2-13B, on a single 80 GB A100 GPU
> |Prune rate|Prune Type|Method|Throughput (tokens/s)|Speedup|Perplexity |
> |---|---|---|---|---|---|
> | 0% | Dense | - | 47.31 | 1.0 | 7.91 |
> | 40% | Unstructured | Wanda | 68.73 | 1.45 | 8.77 |
> | 40% | Low rank & Sparse | Ours | 79.87 | 1.69 | 8.31 |
> | 50% | Unstructured | Wanda | 83.15 | 1.76 | 10.05 |
> | 50% | Semi-structured | Wanda | 92.36 | 1.95 | 16.53 |
> | 50% | Low rank & Sparse | Ours | 102.45 | 2.17 | 9.23 |
>
> ## W2: More justification about Motivation.
> Thanks for your suggestion. We provide a detailed explanation of their rationale and implementation below.
>
> **Motivation of policy gradient.** Our bilevel optimization problem in Eqn. (4) of the paper takes the form of
> $$
> \min_{\theta \in \mathcal{C}} \Phi(\theta) = \mathbb{E}_{(s, \kappa) \sim p(\cdot \mid \theta)} \mathcal{L}(\mathcal{W}(s, \kappa)), \quad \text{with} \quad \mathcal{W}(s, \kappa) = \text{RPCA}(\mathcal{W}, s, \kappa).
> $$
> It involves a nested structure where the outer objective depends on the output of a non-differentiable inner procedure (specifically, RPCA decomposition of weight matrices).
>
>
>
> Existing approaches to bilevel optimization (e.g., [r2, r3]) typically rely on **implicit differentiation**. As discussed in the submission's main text, if we were to follow the conventional approach using implicit differentiation, the resulting gradient estimate would involve second-order derivatives and take the following form:
> $$
> \nabla_{\theta} \Phi(\theta) \approx \nabla_{\theta} \mathcal{W}(s, \kappa) \nabla_{\mathcal{W}} \mathcal{L}(\mathcal{W}(s, \kappa))
> $$
> However, in our case, the RPCA solution $\mathcal{W}(s, \kappa)$ does not admit a tractable or closed-form gradient, making $\nabla_{\theta} \mathcal{W}(s, \kappa)$ cannot be computed directly.
>
> To overcome this challenge, we resort to policy gradient estimator, which allow us to estimate $\nabla_{\theta} \Phi(\theta)$  without calculating $\nabla_{\theta} \mathcal{W}(s, \kappa)$. The PGE is given by such a simpler form:
> $$
> \nabla_ {\theta} \Phi(\theta) = \mathbb{E}_ {(s, \kappa) \sim p(\cdot \mid \theta)} \left[ \mathcal{L}(\mathcal{W}(s, \kappa))\nabla_ {\theta} \log p(s, \kappa \mid \theta) \right].
> $$
> Note that explicitly computing the expectation above is infeasible, we use the policy gradient $\mathcal{L}_ {\mathcal{B}}(\mathcal{W}(s, \kappa))\nabla_{\theta} \log p(s, \kappa \mid \theta)$, which is an unbiased estimation of $\nabla_{\theta} \Phi(\theta)$. **This is why we need and can use this stochastic approximation.**
>
> **Motivation of truncated Gaussion.** The truncated Gaussian prior is our novel design, which effectively mitigates the risk of gradient explosion (see Section 4.2). This issue can arise when using a standard Gaussian distribution, as its gradient contains a $1/\sigma^2$ term. To ensure convergence, one typically needs to reduce $\sigma$ to concentrate the sampling distribution, but this can lead to unstable gradients. The truncated version addresses this problem by allowing us to concentrate samples via truncation while keeping $\sigma$ fixed, thereby maintaining gradient stability.
>
> ## W3: The use of diagonal approximation of $X^\top X$ and More Comparison with HASLLE-free.
> We thank the reviewer for raising this important concern.
>
> **Reasons of using diagonal approximation.** We would like to discuss the reasons why we use diagonal approximation:
> - We adopt this approximation primarily to ensure a fair comparison. Since OATS, the strong baseline in our study, employs this diagonal approximation, we also follow this choice to isolate the superiority of our proposed collaborative allocation framework.
>
> - The core contribution of our work is the collaborative allocation framework, instead of  the specific approximation to $X^\top X$.
>
>
> **Effects of $X^\top X$.** To address the concern, we conduct additional experiments by removing the diagonal approximation used in OATS and directly applying our bilevel framework on the unmodified low-rank sparse structure. The results in **Table R2** below show that our method continues to deliver improvements, indicating that the proposed framework is robust and effective even without relying on this approximation.
>
> #### Table R2 Comparison with OATS on Phi-3 mini 40% prune rate, with and without diag $X^\top X$.
> |Method | PPL | MMLU | zero-shot |
> |---|---|---|---|
> | OATS (w. diag $X^\top X$) | 11.50 | 65.84 | 70.04 |
> | Ours (w. diag $X^\top X$) | 11.03 | 65.92 | 70.56 |
> | OATS (w/o diag $X^\top X$) | 18.34 | 65.31 | 68.22 |
> | Ours (w/o diag $X^\top X$) | 16.48 | 65.73 | 69.14 |
> | Ours(w. full Hassian) | 10.83 | 66.07 | 70.61 |
>
> **Comparisons with HASSLE-free.** According to the reviewer’s suggestion, we compare our method with the HASSLE-free approach. Since HASSLE-free adopts a fixed semi-structured sparsity pattern and a predefined rank, it differs from our adaptive sparsity allocation strategy. As a first step, we reproduced their configuration using our QR-based compression under the same fixed sparsity setting. While this led to slightly lower performance, we argue that such a comparison is not entirely fair, as fixed sparsity allocation is not the main contribution of our method. For a more meaningful evaluation, we further compared both methods under the same overall sparsity ratio (50%), using results directly reported in their original paper. In this setting, our method outperforms HASSLE-free, demonstrating the effectiveness of our adaptive allocation strategy.
>
>
> #### Table R3 Comparison with HASSLE-free on LLaMA3-8B 50% prune rate
> |Method | PPL(wiki) |PIQA|ARC-E|ARC-C|
> |---|---|---|---|---|
> |OATS(3:8+LR) | 11.43 | 75.24 | 65.91 | 39.85 |
> |QR(3:8+LR) | 11.57 | 75.06 | 65.32 | 40.87 |
> |HASSLE-free(3:8+LR) |11.36 |75.79 |67.55|41.04|
> |Ours | 10.18 |78.12 |72.88 |45.42 |
>
> ### References
> [r1] Lujun Li et al. "Discovering Sparsity Allocation for Layer-wise Pruning of Large Language Models."
>
> [r2] Pedregosa. "Hyperparameter optimization with approximate gradient. "
>
> [r3] Grazzi et al. "On the iteration complexity of hypergradient computation."

---

> > ### Comment · Reviewer_nSyU · 2025-08-04
> >
> > Thank you for your response. I have a few further questions.
> >
> > - How are you able to have speed ups for unstructured sparsity, or sparse + low rank? As far as I know this is not supported in vllm, unless I'm missing something obvious. How is sparse + low rank faster than just sparse?
> >
> > - How did you implement the full Hessian version of your method? There are some details over there that are not trivial.

---

> > > ### Author Response · Authors · 2025-08-04
> > >
> > > We sincerely appreciate your active discussion and we would like to address your concerns below.
> > > ## Q1: How are you able to have speed ups for unstructured sparsity, or sparse + low rank?
> > > We achieve speed up for unstructured sparsity based on the sparse matrix computation library such as DeepSparse (for CPU) and ```nm-vllm```(for GPU), which is widely used in recent studies ATP[r1] and OATS. These studies also explicitly demonstrate the performance benefits of sparse matrix computation in their papers. For convenience, we extract the acceleration results in the ATP paper below:
> > > #### Table D1: Inference Speedup results in ATP Paper. LLaMA2-7B with unstructured sparsity.
> > > |Device|Sparsity|Dense|50%|60%|
> > > |---|---|---|---|---|
> > > |CPU|Throughput(tokens/s)|3.40|6.10|7.39|
> > > |CPU|Speedup($\times$)|1.00|1.79|2.17|
> > > |GPU|Throughput|57.29|97.92|111.86|
> > > |GPU|Speedup|1.00|1.71|1.95|
> > >
> > > In their work, the authors evaluated speedup using LLaMA2-7B on an Intel Xeon Silver 4314 CPU with DeepSparse for CPU inference, and on an NVIDIA RTX 4090 GPU using nm-vllm for GPU inference. The corresponding code for both tests is provided in detail on their GitHub repository. For clarity, we excerpt the relevant implementations below:
> > > #### CPU:
> > > - Step 1: install relevant packages sparseml and deepsparse
> > > - Step 2: create sparse model checkpoint, and save to model_path
> > > - Step 3: export the sparse checkpoint to ONNX format
> > > ```bash
> > > sparseml.export --task text-generation model_path
> > > ```
> > > - Step 4: evaluate using deepsparse
> > > ```bash
> > > deepsparse.benchmark model_path/deployment/model.onnx --sequence_length 2048
> > > ```
> > > #### GPU:
> > > - Step1: install relevant packages nm-vllm
> > > - Step2: create sparse model checkpoint, and save to model_path
> > > - Step3: generate outputs with nm-vllm
> > > ```python
> > > from vllm import LLM, SamplingParams
> > > # Example prompt.
> > > prompts =["What is large language model?"]
> > > # Load sparse LLM from model_path.
> > > llm = LLM(model_path, sparsity="sparse_w16a16")
> > > # Generate text from the prompt.
> > > sampling_params = SamplingParams(max_tokens=128)
> > > outputs = llm.generate(prompts, sampling_params=sampling_params)
> > > ```
> > > ## Q2: How is sparse + low rank faster than just sparse?
> > >
> > > To demonstrate the superiority of the sparse+low-rank structure, we provide a general analysis of its acceleration benefits and support our claims with evidence from published works, i.e. OATS, where consistent results on the speedup benefits of sparse+low-rank structure are given.
> > >
> > > **General Analysis of the Speedup Benefits of the Sparse+Low-Rank Structure.** When all methods utilize unstructured sparse matrices with the same prune rate, their acceleration efficiency should be roughly comparable. In contrast, different from the approaches with single sparse structure (e.g., Wanda, SparseGPT), our method adopts a hybrid **sparse + low-rank** structure, which brings two key advantages:
> > >
> > > - Given the same total parameter count, part of our parameters are implemented as low-rank matrices, which are more hardware-friendly and better suited for achieving practical speedups.
> > > - Since the overall number of parameters remains fixed, the sparse component in our method is significantly smaller than that in baseline methods, further contributing to faster computation.
> > >
> > > As a result, our sparse + low-rank design achieves faster inference speed compared to purely sparse counterparts.
> > >
> > > **Consistent Experimental Results in Existing Studies.** We present the table from the OATS paper, which also shows that "sparse+low-rank" can achieve more speed up than "sparse", as shown below:
> > > #### Table D2: Inference Speedup results in OATS Paper. Phi-3 Medium 15B.
> > > |Compression|Method|Throughput(token/s)|Speedup($\times$)|
> > > |---|---|---|---|
> > > |0%|Dense|4.03|1.00|
> > > |40%|Unstructured |5.08 |1.26|
> > > |40%|OATS(Low-rank + Sparse) |6.86 |1.73|
> > > |50%|Unstructured |7.16 |1.78|
> > > |50%|OATS(Low-rank + Sparse)|8.31 |2.06|
> > >
> > > These results were evaluated on a single batch of 2048 tokens on an Intel Xeon Gold 6148 CPU @ 2.40GHz with 32 cores.
> > > ## Q3: How did you implement the full Hessian version of your method?
> > > We would like to address your question as follows.
> > >
> > > In our previous response, motivated by your valuable suggestion, we explored the effect of the full Hessian. Specifically, we follow the computation of the full Hessian as in SparseGPT, i.e., $\text{H} = X^\top X + \lambda \text{I}$, and replace the matrix $\text{D}$ (as introduced in Section 3, which takes form of $\text{D} = \sqrt{\text{diag}(X^\top X)}$) with $\text{D} = 1/\sqrt{\text{diag}(H^{-1})}$ during the inner-loop update. The rest of the inner optimization procedure remains unchanged, consistent with both our original implementation.
> > >
> > > It is worth noting that our experiments are conducted on a relatively small model (Phi-3 mini), which allows us to adopt the standard implementation described above. For larger-scale models, however, computing the inverse of the full Hessian may incur expensive computational overhead.
> > >
> > > We hope this explanation helps clarify your concerns.

---

> > > > ### Comment · Reviewer_nSyU · 2025-08-05
> > > >
> > > > Thank you for your response.
> > > >
> > > > - So if I understand correctly, you are using nm-vllm for sparse GPU inference, which is Neural Magic's fork of vLLM. As far as I can tell this library has not been updated in 9 months or so. This makes this library quite outdated, your dense baselines are not as strong. Additionally, throughput depends heavily on your prompt, prefill length, etc. Are you using a single prompt to calculate the throughput? If yes, that'll be quite inaccurate. How many tokens are you generating?
> > > >
> > > > In the end however, I acknowledge that not everybody knows the details of serving engines, and I appreciate your effort in responding to my comments.
> > > >
> > > > - This, however, does not answer how you get inference acceleration from sparse + low rank. Have you implemented your own kernels? What is the justification for
> > > >
> > > > `Given the same total parameter count, part of our parameters are implemented as low-rank matrices, which are more hardware-friendly and better suited for achieving practical speedups.`?
> > > >
> > > > - Diagonal Hessian: Thank you for clarification. You are still using a diagonal approximation, one that is slightly more accurate.

---

> ### Author Response · Authors · 2025-08-07
>
> We sincerely appreciate your active engagement in the discussion and are happy to further address your concerns.
> ## Q4: nm-vllm is outdated, your dense baselines are not as strong.
> Thank you for your question. We acknowledge that ```nm-vllm``` has indeed not been updated for months. However, it remains one of the few libraries that support GPU-accelerated unstructured sparse inference, and has been adopted in recent research works like ATP[r1]. For these reasons, we have selected it as our benchmark.
>
> We have verified that vLLM indeed has newer versions with better performance. However, the latest version has not yet been integrated into the `nm-vllm` framework. Therefore, using the latest vLLM for dense model inference as a baseline, while evaluating sparse acceleration within `nm-vllm` (which is based on an older vLLM version), leads to an inconsistent setup. As a result, such a setting cannot accurately reflect the actual speedup achieved by sparse acceleration methods.
>
> To the best of our knowledge, there is currently no more appropriate library available for sparse acceleration on GPU. Thus, we think our evaluation setting is a reasonable and feasible choice under the current circumstances.
>
> ## Q5: Are you using a single prompt to calculate the throughput? How many tokens are you generating?
>
> Thank you for your question. We would like to clarify the following: Our testing employed num_requests=16 with a consistent max_token_length setting of 512. All experiments were conducted using the same configurations and were repeated 5 times to ensure fairness and accuracy in our results.
>
> ## Q6: Have you implemented your own kernels?
>
> Currently, there is a lack of GPU inference acceleration libraries specifically designed for low-rank + sparse matrix computation. We did not implement custom kernels from scratch either, although this is theoretically feasible, it would require substantial engineering effort, which is difficult to accomplish during the discussion phase.
>
> Instead, our approach is as follows: we treat the sparse matrix as the backbone and leverage the `nm-vllm` engine to perform sparse inference accordingly. Meanwhile, the low-rank matrices are integrated as LoRA adapters during the forward pass. This design is well-supported by `nm-vllm` and has shown better practical speedup performance.
>
> ## Q7: What is the justification for "Given the same total parameter count, part of our parameters are implemented as low-rank matrices, which are more hardware-friendly and better suited for achieving practical speedups."
>
> Thank you for your question. Our statement is inferred from both our experimental observations and empirical insights from prior work in the field.
>
> Due to the complexity of the ```nm-vllm``` engine, we have not yet conducted a systematic evaluation of the runtime efficiency of each individual operator within the model. However, from the observed acceleration behavior on CPUs, we believe that our assumption, that low-rank matrices offer better hardware efficiency under the same parameter budget, is reasonable.
>
> We will provide a more comprehensive and detailed evaluation in the revised version.
> ## Q8: You are still using a diagonal approximation, one that is slightly more accurate.
> Thank you for your recognition. While this is not the core contribution of our paper, we agree that exploring more accurate inner-loop optimization using the complete full Hessian, beyond diagonal approximations, is a meaningful and promising direction. However, such an extension would require substantial modifications to the algorithm and is difficult to evaluate systematically within the limited discussion phase.
>
> We will conduct a thorough evaluation and include the full results in the revised version.
>
> We hope this explanation helps clarify your concerns.

---

> > ### Comment · Reviewer_nSyU · 2025-08-07
> >
> > Thank you for your response and engagement, I do not have any further questions.

---

> > > ### Author Response · Authors · 2025-08-08
> > >
> > > Thank you very much for your time and thoughtful engagement. We truly appreciate your feedback and the meaningful discussion.

---

### Official Review · Reviewer_sRwV · 2025-06-30

**Clarity:** 3
**Significance:** 4
**Originality:** 3
**Rating:** 5
**Confidence:** 4

**Summary:**

The authors introduce a novel algorithm to compress LLMs via a layer-wise sparse plus low-rank decomposition, using policy gradient to allocate bits across layers (and, within each layer, between the sparse and low rank components).
The algorithm consists of two levels of optimization: an outer level to determine the sparsity percentage and rank for each layer, and an inner level to produce a sparse plus low-rank approximation with the given sparsity and rank.

The inner level of optimization is robust PCA, with an innovation that replaces the computationally-expensive SVD step with a lightweight QR-based approximation.
This is essential for the computational efficiency of the compression algorithm.
The outer level uses policy gradient under a truncated Gaussian model and overall bit budget constraint.
The means of the truncated Gaussians are trainable, with a fixed variance and domain that is gradually reduced over the training process for stability reasons.

Perplexity, task accuracy, and CPU speedup results are presented over several models, with the proposed method overall outperforming baselines.
Some ablation studies and theoretical results provide justification for components of the optimization algorithm.

**Questions:**

(1) How were the values of hyperparameters chosen (e.g., the annealing schedule for the truncated Gaussian support, and the number of inner iterations), and how sensitive is the accuracy of the algorithm to such hyperparameters?
Is there good set of hyperparameters that works across many different LLMs, or are the optimal values expected to be somewhat data-dependent?
Even if they were chosen heuristically, a short statement or discussion would be interesting.

How does the accuracy of RPCA with the QR approximation differ with number of iterations?
A convergence plot for RPCA with and without the QR approximation of SVD could be helpful.

(2) How does the compression time of the algorithm compare to baseline approaches?

(3) How does the empirical sparsity allocation of this method (Figure 4) compare to other sparsity allocation methods like OWL?
Are the differences mainly subtle, or are there clear trends in how this method allocates sparsity between layers?

(4) Instead of diagonally scaling W based on the input activations, is it possible to have a fully calibration-aware loss function, i.e. $||(W - L - S)X||_F^2$ (i.e., is such an algorithm easy to derive)?
Is it known whether this would help downstream accuracy much, compared to the currently-adopted diagonal scaling method?

I don't reasonably see my score decreasing (except if the questions are ignored entirely), and it could increase with very compelling experimental results (a regime or metric where the proposed method does much better than the baselines, such as accuracy under high sparsity levels, or compression time, which a clear corresponding discussion).

**Ethical Concerns:**

["NO or VERY MINOR ethics concerns only"]

**Final Justification:**

The author's rebuttal clarified key points of confusion, namely:
- Which components were original (particularly, the truncated Gaussian distribution) and which were common in the literature,
- Sensitivity of the method to hyperparameters and how the hyperparameters were chosen,
- GPU speedup of the algorithm,
- Ability of the algorithm to be fully calibration-aware.

I have no outstanding concerns. In the context of the rebuttal and the other reviews, I will maintain my score (5). Overall, I think the bilevel optimization framework is principled and an interesting innovation.

**Limitations:**

yes

**Quality:**

3

**Strengths And Weaknesses:**

**Quality**

_Strengths_. The compression algorithm proposed is technically sound and well-polished, and all major components appear to be thoughtfully chosen.
The algorithm is elegant and natural for the problem at hand.
Claims are supported by a broad range of of results, both empirical and theoretical.
All proofs presented are detailed and correct.

Across LLMs and average sparsity levels, the method beats baselines in terms of perplexity, accuracy, and CPU inference speed.
Ablation studies highlight the importance of sparsity allocation, and justify the use of the QR approximation and the inclusion of both sparsity and rank allocation in the outer-level optimization.

Algorithmic details are supported by theoretical derivations, and design choices of the policy gradient level are supported by bias and variance analysis.

_Weaknesses_.
The results section could be improved to better highlight the empirical impact of the paper.
For instance, the benefit in compression time over, e.g., OATS, is briefly touched upon.
That contribution could be emphasized via a small table comparing compression times over the different algorithms compared in Table 1, and should also appear in the introduction.
Also, based on Table 2 (sparsity allocation study), accuracy improvements are more pronounced at higher sparsity levels.
Could Table 1 also include results at 60 and 70\% sparsity?

Speedup on CPU is presented, but no inference latency/bandwidth results on GPU are shown.
Though, since GPU results may require custom CUDA kernels, this is only a minor point.

In addition, the algorithm relies on some hyperparameters (e.g., those of the truncated Gaussian distribution, as well as the number of inner iterations).
Hyperparameter values used for experiments are listed, but it is unclear how sensitive the accuracy of the algorithm is to those parameters (across different models being compressed).
More discussion thereof could be helpful to evaluate ease of practical adoption.

**Clarity**

_Strengths_. The writing is easy to understand, as are the proofs in the appendix.
The structure of the paper makes sense.
The contributions and benefits of the new algorithm are clearly stated, as well as the technical challenges that it solves.

_Weaknesses_.
Some experimental details are missing.
For instance, for Table 3, it is not clear how the allocations are determined for the "w/o Rank," etc., rows of the table.
For the "Integration with other compression methods" section, more background on Wanda and SparseGPT, as well as details on how the proposed algorithm is integrated with those methods, are required for readers not familiar with the literature.

**Significance**

_Strengths_.
The paper provides a novel method for allocating sparsity in sparse + low-rank LLM compression.
Past works have approached this problem heuristically, whereas this one uses a principled optimization framework.
As such, this work has good potential for adoption in future works in the LLM compression community (including outside sparse + low-rank compression; e.g., it would also be useful in the LLM quantization community).
As touched upon. e.g., in the "allocation visualization" section, this work provides insights on the inner workings of LLMs (e.g., the low-rank structure of different layers, as well as which layers store more "dense" data).

_Weaknesses_.
N/A.

**Originality**

_Strengths_.
To my knowledge, the bilevel optimization framework with a policy gradient outer level for LLM compression is novel.
Although all of the components already exist (e.g., the QR approximation of SVD, and use of RPCA for sparse + low-rank decompositions), they are put together in a novel and principled way.
The authors make clear the primary novel contributions of the work.

_Weaknesses_.
To those unfamiliar with some of the literature, it is unclear whether some heuristics, e.g., subtracting a control variate and the annealing schedule of the truncated Gaussian support, are novel or standard practice.

---

> ### Author Rebuttal · Authors · 2025-07-31
>
> Thank you very much for your positive evaluation and thoughtful suggestions. We sincerely appreciate your recognition of our work.
> ## Quality W1: Results presentation.
> We sincerely appreciate your suggestions. Due to space limitation, we present the experimental results under higher sparsity level in our response to reviewer  AQTX's **Table R1**. All other suggestions will be taken in-detail in the revised verision if accepted.
>
> ## Quality W2: GPU results.
> Thanks for your suggestion. In our initial experiments, we followed the evaluation protocols of OATS and OWL, which focus on CPU speed up, and therefore we did not include GPU results.
> To address this, we followed the ATP[r1] protocol and conducted GPU inference tests using `nm-vllm`.
> Due to the space limit, please refer to our response to reviewer AQTX's W6&Q1.
>
> ## Quality W3: Hyperparameter sensitivity.
> Thanks. We would like to clarify following facts to show that our algorithm is not sensitive to these hyperparameters.
> - In fact, we tuned these hyperparameters on Phi-3 mini and applied the same configuration across all other model experiments. Besides, parameters such as the initialization of the truncated Gaussian mean $\mu$ and the number of inner-loop iterations are consistent with those used in OATS.
> - We also conduct ablation studies on these key hyperparameters, and the results are presented in **Table R1** below.
> #### Table R1. Ablation study of hyperparameters Phi3-mini with 50% prune rate, wikitext2 PPL
> |$\sigma^2$ | 1e-3 | 5e-3 | 1e-2 | 5e-2 | 1e-1 |
> |---|---|---|---|---|---|
> | PPL | 15.16 | 14.94 | 14.87 | 14.92 | 15.07 |
> |$\gamma_0$ | **1e-3** | **5e-3** | **1e-2** | **5e-2** | **1e-1** |
> | PPL | 15.22 | 15.01 | 14.93 | 14.87 | 14.99 |
> |$\gamma_{target}$ | **1e-3** | **5e-3** | **1e-2** | **5e-2** | **1e-1** |
> | PPL | 14.88 | 14.87 | 14.90 | 15.07 | 15.18 |
>
> ## Clarity W1: Experimental details.
> We sincerely apologize for the omissions in the original submission and appreciate your insightful comments. We will ensure that these aspects are addressed thoroughly in the final version.
>
> First, regarding Table 3(in submission), our goal is to isolate the effect of different components in our method—namely, sparsity allocation across layers and rank ratio across matrices. Specifically:
>
> * **“w/o Rank”** refers to **not performing learnable rank allocation**; each matrix is assigned a fixed rank ratio, while the layer-wise sparsity is still learned.
> * **“w/o Sparsity”** denotes the setting where **compression rates are fixed uniformly across layers**, and only the rank ratios of matrices are learned.
> * **“w/o Sparsity & Rank”** disables both learnable sparsity and rank allocation. All layers and matrices are compressed with uniform configurations.
>
> Second, we agree that more background on Wanda and SparseGPT would be beneficial, especially for readers less familiar with the literature.  Specifically, we replaced the inner optimization in our bilevel framework—from an RPCA-based method using QR decomposition—to Wanda and SparseGPT, enabling the outer loop to learn the sparsity allocation for each layer. We will add a paragraph to introduce these methods in the revised version if accepted.
>
> ## Originality W1: Heuristics' originality.
> We apologize for not clarifying the originality and standardness of some of our design choices. We will include following clarifications in the revised version if accepted.
>
> - First, subtracting a zero-mean and highly correlated control variate is a core idea in variance reduction techniques used in statistics and Monte Carlo methods. However, the specific design of variance reduction strategies can vary significantly across different applications. Our contribution lies in tailoring a variance reduction approach specifically for our bilevel optimization framework, which enables more stable policy gradient estimation.
> - Second, the truncated Gaussian prior is our novel design, which effectively mitigates the risk of gradient explosion (see Section 4.2). This issue can arise when using a standard Gaussian distribution, as its gradient contains a $1/\sigma^2$ term. To ensure convergence, one typically needs to reduce $\sigma$ to concentrate the sampling distribution, but this can lead to unstable gradients. The truncated version addresses this problem by allowing us to concentrate samples via truncation while keeping $\sigma$ fixed, thereby maintaining gradient stability.
> - Third, the cubic annealing schedule has been used in other applications, such as control the prune rate during training [r2]. We introduce it to our truncated Gaussian prior and find it works well.
>
>
> ## Q1: Hyperparameters and RPCA convergence.
> Regarding hyperparameters' sensitivity, please refer to our response to Quality W3.
>
> Unlike conventional RPCA solvers that rely on accurate decomposition, our bilevel pruning framework only requires approximate gradients. The QR-based estimator strikes a balance between efficiency and performance. Empirical results confirm that this choice does not hinder optimization quality.
> #### Table R2. Relative error(%) of q_proj and o_proj in Phi-3 mini's 1st Layer. The matrix o_proj has relative higher rank, leading to larger approximation error for both SVD and QR.
> |Matrix|Method | 1 |5 |20 |40 |80 | 120|160 |Total Time(s)|
> |---|---|---|---|---|---|---|---|---|---|
> |q_proj|SVD | 3.62| 2.94|2.63 |2.54 |2.49 |2.47 | 2.46 |  126.97  |
> |q_proj|QR |14.7 |7.17| 4.33| 3.50| 3.02|2.83 |2.69 |  0.71s |
> |o_proj|SVD | 14.37 | 11.96 | 10.83 | 10.53 | 10.36 | 10.31 | 10.30 |119.88s  |
> |o_proj|QR  | 15.69 | 13.36 | 11.78 | 11.13 | 10.91 | 10.75 | 10.68 | 0.69s |
> ## Q2: Compression time.
> We acknowledge that, unlike one‑shot methods such as Wanda or SparseGPT, our bilevel framework incurs additional runtime. However, our entire process still completes within a few hours on standard hardware, which is always affordable in downstearm applications.
>
> Importantly, model compression is a one‑time offline cost, where **peak GPU memory usage often outweighs total runtime in practical deployments. Our method uses no more memory than one‑shot approaches but delivers superior accuracy and inference speedups**, making the extra time investment worthwhile.
> The results of compression time are summarized in **Table R3**. Even **QR**, a simplified version of our method, still outperforms SparseGPT and Wanda under comparable time cost.
>
> #### Table R3. Compression time, Phi-3 mini 60% prune rate
> |Method | Prune Time | Wiki | MMLU | Zero-shot |
> |---|---|---|---|---|
> |SparseGPT | 27min | 51.44 | 37.37 | 54.61 |
> |Wanda | 12min | 52.28 | 36.96 | 55.48 |
> |OATS | 5h | 42.37 | 43.96 | 57.82 |
> |QR | 15min |48.8  | 38.22 | 56.01 |
> |Ours | 4.5h | 32.46 | 52.27 | 59.24 |
>
> ## Q3: Sparsity allocation compared with OWL.
> Due to the rebuttal constraint, we are unable to present the full comparison figure. Instead, we report the sparsity allocations of several selected layers in the table below. Compared with OWL, our method exhibits a similar overall trend—allocating more parameters to the early layers, applying stronger compression to the middle-to-late layers, and preserving more parameters in the final layer. This aligns with the general understanding of layer-wise redundancy in large language models. Notably, our method shows larger differences than OWL, more accurately reflecting the redundancy across layers and contributes to better overall performance.
> #### Table R4. Sparsity allocation OWL & Ours Phi-3 mini 60% prune rate
> |Method\Layer | 0 | 6 | 12 | 18 | 24 | 31| PPL |
> |---|---|---|---|---|---|---| ---|
> |OWL | 0.519 | 0.537 | 0.560 | 0.630 | 0.669 | 0.595 | 35.37|
> |Ours | 0.544 | 0.518 | 0.553 | 0.618 | 0.690 | 0.563 | 32.46 |
> ## Q4: Fully calibration-aware Optimization.
> Thank you very much for your insightful suggestion regarding the improvement of our algorithm. After careful derivation, we confirm that the calibration-aware loss function you proposed can indeed be formulated and optimized. We present the derivation in detail below.
>
> Here is a brief derivation of the refined optimization problem:
> $$
> \begin{aligned}
>     & min_ {U,V,S}  \lVert(W- UV-S)X\rVert_ F^2,  \\\\
>     &\text{s.t.} ~\text{rank}(UV) \leq r, ~\lVert S\rVert_ 0 \leq K. \\\\
> \end{aligned}
> $$
> We still use alternating minimization strategy. In each iteration, we update $U$, $V$, and  $S$ as follows:
> $$
> \begin{cases}
>     U_{t+1} = B_{t+1} V_t^\top (V_t A V_t^\top)^{-1},\\\\
>     V_{t+1} = (U_{t+1}^\top U_{t+1})^{-1} U_{t+1}^\top (B_{t+1}A^{-1}),\\\\
>     S = \mathcal{P_C}(W-U_{t+1}V_{t+1}),\\
> \end{cases}
> $$
> where $A=XX^\top, B_{t+1}=(W-S_t)XX^\top$.
> Based on the derivation in the main text, we obtain the following update rules using QR decomposition:
> $$
> \begin{cases}
>     Q_t R_t = B_{t+1}V_i^\top,\\\\
>     U_{t+1} = Q,\\\\
>     V_{t+1} = Q^\top(B_{t+1}A^{-1}).
> \end{cases}
> $$
>
> We conduct experiment to evaluate this approach and report the results in **Table R5**. It shows that performance of these two methods are comparable. If the paper is accepted, we will include additional experiments in the revised version to evaluate the superiority of calibration-aware method.
> #### Table R5. Experiment on Phi-3 mini 50% prune rate
> | Method  | PPL |MMLU | zero-shot |
> | --- | ---  | --- | --- |
> | Ours | 14.87 | 60.57 | 69.37|
> | Calibration-aware | 14.79 | 60.76  | 69.53 |
>
> ### References
> [r1] Lujun Li et al. "Discovering Sparsity Allocation for Layer-wise Pruning of Large Language Models."
>
> [r2] Zhu, M. H. et al. "To Prune, or Not to Prune: Exploring the Efficacy of Pruning for Model Compression."

---

> > ### Comment · Reviewer_sRwV · 2025-08-04
> >
> > Thank you for your thoughtful clarifications and additional results. I don't have any more questions at this time.

---

> > > ### Author Response · Authors · 2025-08-04
> > >
> > > Thank you for your response and encouraging feedback on our work. If you have any further questions or suggestions, we would be glad to continue the discussion at your convenience.

---

### Official Review · Reviewer_ZdES · 2025-07-01

**Clarity:** 3
**Significance:** 2
**Originality:** 3
**Rating:** 4
**Confidence:** 4

**Summary:**

This paper proposes a bilevel optimization framework that jointly integrates learning and matrix approximation for compressing large language models. The work focuses on developing a pruning methodology for LLMs with comprehensive evaluation protocols to assess the performance of pruned models. While the technical details are not fully available in the provided context, the paper appears to present a systematic approach to model compression that aims to reduce the computational requirements of LLMs while maintaining their performance through an integrated optimization strategy.

**Questions:**

1. Scalability concerns: How does the proposed bilevel optimization framework scale with model size? Has it been tested on models with 10B+ parameters, and what are the memory and computational requirements during the compression process?

2. Comparison with one-shot methods: How does the proposed method compare with recent one-shot pruning methods like SparseGPT in terms of both compression quality and time required for compression?

3. Generalization across architectures: Has the method been evaluated across different LLM architectures (e.g., GPT, LLaMA, BLOOM), and does it require architecture-specific tuning?

4. Fine-tuning requirements: Does the compressed model require task-specific fine-tuning to maintain performance, or can it maintain zero-shot capabilities after compression?

5. Hardware efficiency: Have the authors evaluated the actual inference speedup on different hardware platforms (GPUs, CPUs, edge devices), considering that theoretical compression doesn't always translate to practical speedup?

**Ethical Concerns:**

["NO or VERY MINOR ethics concerns only"]

**Final Justification:**

The author resolved my confusion and I will maintain my score.

**Limitations:**

yes

**Quality:**

3

**Strengths And Weaknesses:**

**Strengths**

1. Novel optimization framework: The proposed bilevel optimization approach that jointly considers learning and matrix approximation represents a theoretically principled method for model compression, potentially offering advantages over traditional sequential compression approaches.

2. Comprehensive evaluation: The paper mentions providing "comprehensive documentation" on evaluation protocols, suggesting thorough experimental validation of the proposed method across multiple metrics and benchmarks.

3. Technical innovation: The authors claim "a series of technical innovations" in their method section, indicating potential contributions beyond the basic framework that could advance the state-of-the-art in LLM compression.

**Weaknesses**

1. Limited recent citations: The paper appears to lack references to several recent important works in LLM compression and pruning. Recent papers that should be cited include:

    a) "GPTQ: Accurate Post-Training Quantization for Generative Pre-trained Transformers" (Frantar et al., ICLR 2023)

    b) "Outlier Suppression: Pushing the Limits of Low-bit Transformer Language Models" (Wei et al., NeurIPS 2022)

    c) AffineQuant: Affine Transformation Quantization for Large Language Models (Ma et al., ICLR 2024)

2. Unclear computational overhead: Without access to the main content, it's unclear whether the bilevel optimization framework introduces significant computational overhead during the compression process, which could limit its practical applicability to very large models.

3. Limited discussion of trade-offs: The available context doesn't indicate whether the paper thoroughly discusses the trade-offs between compression ratio, model performance, and computational efficiency during inference.

---

> ### Author Rebuttal · Authors · 2025-07-31
>
> ## W1: Limited citations on quantization.
> Thank you for this valuable suggestion. We fully agree that incorporating these recent works in the area of LLM compression will strengthen the contextualization and relevance of our work. We will include a paragraph reviewing these papers in the revised version if accepted.
>
> ## W2 & Q1: Computational overhead and Scability.
>  Actually, our method has good performance on scalability due to the following reasons:
>
> 1. **On memory cost.** Since our pruning process does not require backpropagation, the computations can be performed in a layer-wise manner. Specifically, we load and process the model one layer at a time, which effectively prevents memory usage from scaling with the overall model size.
>
> 2. **On computational cost.** Thanks to the efficiency of our QR decomposition—over 75× faster than SVD—the computational cost for the inner loop is of the same order as a single forward pass. In all our experiments, we set the number of epochs to 1, making the overall cost affordable for most downstream practitioners, even when working with large-scale LLMs.
>
> **Results on Time Cost.** **Table R2** reports the time costs for QR decomposition, and other time (model (layer) loading, and GPU forward computation, etc.) per layer. The results show that these costs scale approximately linearly with the model size. This verifies our analysis above. Notice that loading and forward occupies most of the time cost, which can be improved by  engineering optimization techniques such as DeepSpeed.
> #### Table R1. Time taken of QR-based RPCA and other time
> |Model Size | RPCA time | Other time | GPU cost|
> |---|---|---|---|
> |Phi-3 mini(3.8B) | 3.39 | 22.12 | 14.28G |
> |LLaMA2-7B | 5.86 | 43.71 | 18.58G |
> |LLaMA2-13B | 11.29 | 63.89 | 22.86G |
> |LLaMA3-70B | 21.46 | 117.31 | 55G |
>
> **Results on LLaMA3-70B.** Table R2 shows that our method can outperform  the baseline OATS. More results will be included  in the revised version if accepted.
>
> #### Table R2. Results on  LLaMA3-70B.
> |Method|Time|PPL|MMLU|zero-shot|
> |:-:|:-:|:-:|:-:|:-:|
> |Dense|-|4.03|76.67|75.28|
> |OATS|120h|4.95 |74.02 |73.41|
> |Ours|80h|4.67|75.13|73.79|
>
> ## W3: Limited discussion of trade-offs.
> We thank the reviewer for pointing out the lack of discussion on the trade-offs. In the revised version, we will include an additional section that explicitly analyzes the relationship among these factors.
>
> Specifically, based on our findings and prior work, we observe that under sparsity levels below 50%, the compressed model retains most of its original performance while achieving meaningful acceleration on both CPU and GPU platforms. When the sparsity increases beyond this point, although the inference speed continues to improve, the performance degradation becomes significant. We suggest that a sparsity level around 50% strikes a favorable balance between model efficiency and performance in practical scenarios.
>
>
> ## Q2: Comparison with one-shot methods.
> We acknowledge that, unlike one‑shot methods such as Wanda or SparseGPT, our bilevel framework incurs additional runtime. However, our entire process still completes within a few hours on standard hardware, which is always affordable in downstearm applications.
>
> Importantly, model compression is a one‑time offline cost, where **peak GPU memory usage often outweighs total runtime in practical deployments. Our method uses no more memory than one‑shot approaches but delivers superior accuracy and inference speedups**, making the extra time investment worthwhile.
> The results of compression time are summarized in **Table R3**. Even **QR**, a simplified version of our method, still outperforms SparseGPT and Wanda under comparable time cost.
>
> #### Table R3. Compression time, Phi-3 mini 60% prune rate
> |Method | Prune Time | Wiki | MMLU | Zero-shot |
> |---|---|---|---|---|
> |SparseGPT | 27min | 51.44 | 37.37 | 54.61 |
> |Wanda | 12min | 52.28 | 36.96 | 55.48 |
> |OATS | 5h | 42.37 | 43.96 | 57.82 |
> |QR | 15min |48.8  | 38.22 | 56.01 |
> |Ours | 4.5h | 32.46 | 52.27 | 59.24 |
>
> ## Q3: Generalization across architectures.
> Our main experiments focus on Phi and LLaMA2/3 models. Since our method performs optimization at the matrix level without relying on specific model architectures, it is generally applicable to other network structures as well. To further address the reviewer’s concern, we have conducted additional experiments on Qwen2.5-3B, with results shown in the table below.
>
> #### Table R4. Additional experiments on Qwen2.5 3B
> |Model | Prune rate | Wiki | MMLU | Zero-shot |
> |---|---|---|---|---|
> |Qwen2.5-3B | 0% | 11.02 | 65.99 | 68.49 |
> |Qwen2.5-3B | 50%(Ours) | 14.41 | 59.13 | 66.05 |
>
> ## Q4: Fine-tuning requirements.
> To ensure a fair comparison aligned with current research paradigms, we did not perform any task-specific fine-tuning for any of the compression methods and instead evaluated their zero-shot performance. We would like to emphasize two points:
> - First, at a sparsity level of 50%, the compressed models are able to retain over 95% of their zero-shot performance, making fine-tuning unnecessary.
> - Second, fair comparisons under fine-tuning settings are non-trivial. While our method can efficiently fine-tuning compressed model using low-rank component, methods like Wanda and SparseGPT require careful structural modifications, which fall beyond the scope of this work.  We leave this direction for future exploration.
> ## Q5: Hardware efficiency.
> We appreciate the reviewer’s comment and agree that presenting inference speedup over different platforms is important. In our initial experiments, we followed the evaluation protocols of OATS and OWL, which focus on CPU speed up, and therefore we did not include GPU results.
> To address this, we followed the ATP[r1] protocol and conducted GPU inference tests using `nm-vllm`. The results are presented in **Table R5**.
> Compared with unstructured methods, our approach, Low rank & Sparse, leads to more practically meaningful GPU inference acceleration.
> #### Table R5. GPU inference speedup LLaMA2-13B, on a single 80 GB A100 GPU
> |Prune rate|Prune Type|Method|Throughput (tokens/s)|Speedup|Perplexity |
> |---|---|---|---|---|---|
> | 0% | Dense | - | 47.31 | 1.0 | 7.91 |
> | 40% | Unstructured | Wanda | 68.73 | 1.45 | 8.77 |
> | 40% | Low rank & Sparse | Ours | 79.87 | 1.69 | 8.31 |
> | 50% | Unstructured | Wanda | 83.15 | 1.76 | 10.05 |
> | 50% | Semi-structured | Wanda | 92.36 | 1.95 | 16.53 |
> | 50% | Low rank & Sparse | Ours | 102.45 | 2.17 | 9.23 |
>
> ### References
> [r1] Lujun Li et al. "Discovering Sparsity Allocation for Layer-wise Pruning of Large Language Models."

---

> ### Author Response · Authors · 2025-08-01
>
> Thank you very much for your response and encouraging feedback on our work. Do you have any further questions or comments? We would be more than happy to engage in further discussion at your convenience.

---

> > ### Comment · Reviewer_ZdES · 2025-08-07
> >
> > Thank you very much for the clarifications. I have no further questions and will maintain my original score.

---

> > > ### Author Response · Authors · 2025-08-08
> > >
> > > Thank you very much for your positive feedback. If you have any further questions or concerns, we would be more than happy to address them.

---

### Official Review · Reviewer_tY2u · 2025-07-03

**Clarity:** 3
**Significance:** 3
**Originality:** 3
**Rating:** 5
**Confidence:** 3

**Summary:**

This paper proposes a compression method for LLMs using a bilevel optimization framework that integrates learning with sparse and low-rank matrix approximations. The outer loop optimizes sparsity allocation using truncated Gaussian priors and policy gradients, while the inner loop efficiently solves matrix approximation problems via QR decomposition instead of conventional SVD. Experiments demonstrate competitive performance in accuracy preservation under significant sparsity.

**Questions:**

How sensitive is the method to variations in critical hyperparameters like variance and truncation?

Are there theoretical insights or proofs available regarding the convergence of your bilevel optimization?

Is the any indication that the proposed method can scale efficiently to models significantly larger than those tested?

**Ethical Concerns:**

["NO or VERY MINOR ethics concerns only"]

**Final Justification:**

The authors have sufficiently addressed my initial concerns in their rebuttal. I have updated my score accordingly.

**Limitations:**

yes

**Paper Formatting Concerns:**

Formatting is appropriate, clear, and complies with NeurIPS guidelines.

**Quality:**

3

**Strengths And Weaknesses:**

[Strengths]
The integration of bilevel optimization and truncated Gaussian-based policy gradients seems novel.

They demonstrate improvements over several existing methods in terms of accuracy and speed on multiple popular LLM architectures such as Phi-3, Llama2-7B, Llama2-13B, and Llama3-8B.

[Weaknesses]
Theoretical convergence guarantees, optimality conditions, and stability analyses of the proposed method are not explored sufficiently.

Success heavily dependent on several hyperparameters like variance and truncation parameters, which may require extensive tuning and limit practical utility.

Although QR decomposition is more efficient than SVD, the method remains computationally demanding for large-scale applications.

Practical deployment may be hindered due to extensive hyperparameter tuning requirements.

---

> ### Author Rebuttal · Authors · 2025-07-31
>
> ## W1: Theoretical guarantees, optimality conditions and so on.
> Thank you for pointing out this limitation. We provide a theoretical convergence analysis with optimality conditions in our response to your Question 2. We will incorporate these discussions into the revised version if accepted.
>
> ## W2, W4 & Q1: Hyperparameters' sensitivity.
> We acknowledge that our bilevel framework introduces several hyperparameters. However, we would like to clarify following facts to show that our algorithm is not sensitive to these hyperparameters.
> - In fact, we tuned these hyperparameters on Phi-3 mini and applied the same configuration across all other model experiments. Besides, parameters such as the initialization of the truncated Gaussian mean $\mu$ and the number of inner-loop iterations are consistent with those used in OATS.
> - We also conduct ablation studies on these key hps, and the results are presented in **Table R1** below.
> - Moreover, theoretical analysis (see our response to your Question 2) shows the dependency of convergence on the parameters such as $\gamma$.
> #### Table R1. Ablation study of hyperparameters Phi3-mini with 50% prune rate, wikitext2 PPL
> |$\sigma^2$ | 1e-3 | 5e-3 | 1e-2 | 5e-2 | 1e-1 |
> |---|---|---|---|---|---|
> | PPL | 15.16 | 14.94 | 14.87 | 14.92 | 15.07 |
> |$\gamma_0$ | **1e-3** | **5e-3** | **1e-2** | **5e-2** | **1e-1** |
> | PPL | 15.22 | 15.01 | 14.93 | 14.87 | 14.99 |
> |$\gamma_{target}$ | **1e-3** | **5e-3** | **1e-2** | **5e-2** | **1e-1** |
> | PPL | 14.88 | 14.87 | 14.90 | 15.07 | 15.18 |
> ## W3 & Q3: Large-scale applications.
> Thank you for pointing out this concern. Due to the space limit, please refer to our response to reviewer ZdES's W2 & Q1.
>
> ## Q2  Theoretical analysis:
> We would like to deliver a formal form of our theorem below by adding some assumptions following [r1].
>
> **Theorem** For bilevel problem with the form as:
> $$
> \min_ {\theta \in \mathcal{C}} \Phi(\theta) = \mathbb{E}_ {(s, \kappa) \sim p(\cdot \mid \theta)} \mathcal{L}(\mathcal{W}(s, \kappa)) \quad \text{with} \quad \mathcal{W}(s, \kappa) = \text{RPCA}(\mathcal{W}, s, \kappa).
> $$
> We assume $\Phi(\theta)$ is $L$-smooth, and the policy gradient variance satisfies:
>
> $$
> \mathbb{E}\left\lVert\mathcal{L}(\mathcal{W}(s, \kappa))\nabla_{\theta} \log p(s, \kappa \mid \theta)-\nabla_{\theta} \Phi(\theta) \right\rVert^2 \leq \tilde{\sigma}^2.
> $$
>
> Let $\eta < 1/L$ and define the gradient mapping $\mathcal{G}^t$ at iteration $t$ as:
>
> $$
> \mathcal{G}^t = \frac{1}{\eta}\left(\theta^t - \mathcal{P}_{\mathcal{C}}\left(\theta^t - \eta \nabla _{\theta} \Phi(\theta^t)\right)\right),
> $$
>
> then when $T \rightarrow \infty$, we have:
>
> $$
> \frac{1}{T} \sum_{t=1}^T \mathbb{E} \lVert \mathcal{G}^t\rVert^2 \leq \frac{8 - 2L\eta}{2 - L\eta} \tilde{\sigma}^2.
> $$
>
> **Remark** We would like to point out the following:
>
> - We give this theorem just to show that our algorithm behaves similarly to general projected/proximal stochastic gradient descent algorithms for one-level optimization problems (e.g., [r2]), rather than to characterize how fast our algorithm converges. Therefore, we do not consider variance reduction techniques, which are out of scope for this work.
>
> - $\tilde{\sigma}^2$ is the variance of our gradient estimator PGE with a single sampling of (s,$\kappa$). It can be reduced using multiple sampling, larger batch sizes and other variance reduction techniques, hence the LHS of Equation (1) can converge to a small value.
>
> -  As shown in line 912 of Appendix D.3, $\tilde{\sigma}^2$ becomes small when $\gamma$ is small. In our experiments, $\gamma$ is gradually reduced to a smaller value to facilitate convergence.
> ---
> We need the following lemmas about the properties of the projection operator, which can be found in [r2].
>
> **Lemma 1 (Firmly Nonexpansive Operators)**
> Let $\mathcal{C} \subset \mathbb{R}^d$ be compact and convex. For any $\mathbf{u}, \mathbf{v} \in \mathbb{R}^d$:
>
> $$
> \lVert \mathcal{P}_ {\mathcal{C}}(\mathbf{u}) - \mathcal{P}_ {\mathcal{C}}(\mathbf{v}) \rVert^2 \leq (\mathbf{u} - \mathbf{v})^\top \left( \mathcal{P}_ {\mathcal{C}}(\mathbf{u}) - \mathcal{P}_ {\mathcal{C}}(\mathbf{v}) \right).
> $$
>
> **Lemma 2**
> Let $\mathcal{C} \subset \mathbb{R}^d$ be compact and convex. Then for any $\mathbf{c} \in \mathcal{C}$ and $\mathbf{u}, \mathbf{v} \in \mathbb{R}^d$:
>
> $$
> \lVert\mathcal{P}_ {\mathcal{C}}(\mathbf{c} + \mathbf{u}) - \mathcal{P}_ {\mathcal{C}}(\mathbf{c} + \mathbf{v}) \rVert \leq \lVert \mathbf{u} - \mathbf{v} \rVert.
> $$
>
> ---
> **Proof of Theorem:**
> We define:
>
> $$
> \mathbf{g}^t = \mathcal{L}_ {\mathcal{B}}(\mathcal{W}(s, \kappa)) \nabla_ {\theta} \log p(s, \kappa| \theta^t).
> $$
>
> Our method updates $\theta$ as:
>
> $$
> \theta^{t+1} = \mathcal{P}_{\mathcal{C}}\left( \theta^t - \eta \mathbf{g}^t \right).
> $$
>
> Let the stochastic and deterministic gradient mappings be
>
> $$
> \begin{aligned}
>     \hat{\mathcal{G}}^t &= \frac{1}{\eta}\left(\theta^{t}-\mathcal{P}_ {\mathcal{C}}\left(\theta^t- \eta \mathbf{g}^t\right)\right) = \frac{1}{\eta}\left(\theta^{t}-\theta^{t+1}\right), \\\\
>     \mathcal{G}^t &= \frac{1}{\eta}\left(\theta^{t}-\mathcal{P}_ {\mathcal{C}}\left(\theta^t- \eta \nabla\Phi(\theta^t)\right)\right),
> \end{aligned}
> $$
>
> we can have
>
> $$
> \begin{aligned}
>     \Phi(\theta^{t+1}) &\leq \Phi(\theta^{t}) + \langle \nabla\Phi(\theta^{t}), \theta^{t+1}-\theta^t \rangle + \frac{L}{2} \lVert \theta^{t+1}-\theta^t \rVert^2 \\\\
>     &= \Phi(\theta^{t}) - \eta \langle \nabla\Phi(\theta^{t}), \hat{\mathcal{G}}^t \rangle + \frac{L\eta^2}{2} \lVert \hat{\mathcal{G}}^t \rVert^2 \\\\
>     &= \Phi(\theta^{t}) - \eta \langle \nabla\Phi(\theta^{t}) - \mathbf{g}^t + \mathbf{g}^t, \hat{\mathcal{G}}^t \rangle + \frac{L\eta^2}{2} \lVert \hat{\mathcal{G}}^t \rVert^2 \\\\
>     &= \Phi(\theta^{t}) - \eta \langle \mathbf{g}^t, \hat{\mathcal{G}}^t \rangle + \frac{L\eta^2}{2} \lVert \hat{\mathcal{G}}^t \rVert^2 + \eta \langle \delta^t, \hat{\mathcal{G}}^t \rangle \quad \text{(here } \delta^t = \mathbf{g}^t - \nabla\Phi(\theta^{t})) \\\\
>     &\leq \Phi(\theta^{t}) - \eta \lVert \hat{\mathcal{G}}^t \rVert^2 + \frac{L\eta^2}{2} \lVert \hat{\mathcal{G}}^t \rVert^2 + \eta \langle \delta^t, \hat{\mathcal{G}}^t \rangle  \quad \quad \text{(Lemma 1)}\\\\
>     &\leq \Phi(\theta^{t}) - (\eta - \frac{L\eta^2}{2}) \lVert \hat{\mathcal{G}}^t \rVert^2 + \eta \langle \delta^t, \hat{\mathcal{G}}^t \rangle \\\\
>     &= \Phi(\theta^{t}) - (\eta - \frac{L\eta^2}{2}) \lVert \hat{\mathcal{G}}^t \rVert^2 + \eta \langle \delta^t, \mathcal{G}^t \rangle + \eta \langle \delta^t, \hat{\mathcal{G}}^t - \mathcal{G}^t \rangle \\\\
>     &\leq \Phi(\theta^{t}) - (\eta - \frac{L\eta^2}{2}) \lVert \hat{\mathcal{G}}^t \rVert^2 + \eta \langle \delta^t, \mathcal{G}^t \rangle + \eta \lVert \delta^t \rVert \lVert \hat{\mathcal{G}}^t - \mathcal{G}^t \rVert \\\\
>     &\leq \Phi(\theta^{t}) - (\eta - \frac{L\eta^2}{2}) \lVert \hat{\mathcal{G}}^t \rVert^2 + \eta \langle \delta^t, \mathcal{G}^t \rangle + \eta \lVert \delta^t \rVert^2. \quad \quad (\text{Lemma 2})
> \end{aligned}
> $$
>
> Therefore, we can get
>
> $$
> (\eta - \frac{L\eta^2}{2}) \lVert \hat{\mathcal{G}}^t \rVert^2 \leq \Phi(\theta^{t}) - \Phi(\theta^{t+1}) + \eta \langle \delta^t, \mathcal{G}^t \rangle + \eta \lVert \delta^t \rVert^2.
> $$
>
> Thus, we obtain
>
> $$
> \sum_{t=1}^T (\eta - \frac{L\eta^2}{2}) \lVert \hat{\mathcal{G}}^t \rVert^2 \leq \Phi(\theta^{1}) - \Phi(\theta^{T+1}) + \eta \sum_{t=1}^T \left( \langle \delta^t, \mathcal{G}^t \rangle + \lVert \delta^t \rVert^2 \right). \tag{1}
> $$
>
> Now, we turn to analyze the term $\langle \delta^t, \mathcal{G}^t \rangle$ as follows:
>
> $$
> \mathbb{E} \langle \delta^t, \mathcal{G}^t \rangle = \mathbb{E}_ {\theta^t} \mathbb{E}_ {\cdot | \theta^t} \left( \langle \mathbf{g}^t - \nabla\Phi(\theta^t), \mathcal{G}^t \rangle \mid \theta^t \right) = 0. \tag{2}
> $$
>
> For $\lVert \delta^t \rVert^2$, we have
>
> $$
> \mathbb{E} \lVert \delta^t \rVert^2 = \mathbb{E} \lVert \mathbf{g}^t - \nabla\Phi(\theta^{t}) \rVert^2 \leq \tilde{\sigma}^2. \tag{3}
> $$
>
> Combining inequalities (2), (1), and (3), we have
>
> $$
> \frac{1}{T} \sum_{t=1}^T \mathbb{E} \lVert \hat{\mathcal{G}}^t \rVert^2 \leq \frac{\Phi(\theta^1) - \Phi^*}{(1 - L\eta/2)T} + \frac{\tilde{\sigma}^2}{1 - L\eta/2}. \tag{4}
> $$
>
> Finally, we bound $\mathbb{E} \lVert \mathcal{G}^t \rVert^2$ as follows:
>
> $$
> \begin{align*}
>     \mathbb{E} \lVert \mathcal{G}^t \rVert^2 &\leq 2 \mathbb{E} \lVert \hat{\mathcal{G}}^t \rVert^2 + 2 \mathbb{E} \lVert \mathbf{g}^t - \nabla \Phi(\theta^t) \rVert^2 \\\\
>     &\leq 2 \mathbb{E} \lVert \hat{\mathcal{G}}^t \rVert^2 + 2 \tilde{\sigma}^2. \tag{5}
> \end{align*}
> $$
>
> Combining inequalities (5) and (4), as $T \rightarrow \infty$, we obtain
>
> $$
> \frac{1}{T} \sum_ {t=1}^T \mathbb{E} \lVert \mathcal{G}^t \rVert^2 \leq \frac{2}{1 - L\eta/2} \left( \frac{\Phi(\theta^1) - \Phi^*}{T} + (2 - L\eta/2)\tilde{\sigma}^2 \right) \rightarrow \frac{8 - 2L\eta}{2 - L\eta} \tilde{\sigma}^2.
> $$
>
> $\blacksquare$
>
> ### References
> [r1] Pedregosa et al., "Hyperparameter optimization with approximate gradient."
>
> [r2] Ghadimi and Lan, "Mini-batch stochastic approximation methods for nonconvex stochastic composite optimization."

---

### Official Review · Reviewer_AQTX · 2025-07-09

**Clarity:** 3
**Significance:** 3
**Originality:** 2
**Rating:** 4
**Confidence:** 3

**Summary:**

Authors propose a novel optimization framework based on Robust PCA for the compression of LLMs. They combine learning-based sparsity allocation and QR-based matrix approximation algorithms. There are two technical contributions in the paper: (1) a truncated Gaussian prior-based probabilistic parameterization with policy gradient estimator for the outer loop optimization, and (2) a QR-based matrix approximation algorithm for efficient inner loop computation. The approach automatically learns layer-wise sparsities and matrix-wise retained ranks, demonstrating strong performance on Phi-3 and Llama model families.

**Questions:**

1. All speedup evaluations are on CPU using DeepSparse. Given that most LLM inference happens on GPUs, can you provide GPU inference benchmarks? How does the structured sparsity from RPCA translate to GPU acceleration?
2. Can you provide the convergence analysis of the proposed method? How does different hp affect the convergence rate?
3. There are multiple hyperparameters, how do they affect the results? I feel like it will be really difficult to handle these many hyperparameters.
4. Time taken for bilevel optimization compared to one-shot methods like Wanda/SparseGPT?

**Ethical Concerns:**

["NO or VERY MINOR ethics concerns only"]

**Final Justification:**

Based on author response and remarks from other reviewers, I have decided to retain the score.

**Limitations:**

The authors appropriately discuss some of the limitations in the appendix, including computational overhead and hyperparameter sensitivity. Additional limitations include scalability to very large models and the need for careful tuning of different hyperparameters.

**Quality:**

3

**Strengths And Weaknesses:**

Strengths:
1. The paper is built on a strong technical and theoretical foundation. Even though RPCA has been used for model pruning; problem formulation, methodology and optimization framework proposed in the paper is novel.
2. Authors have done proper analysis of all the theorems and provided the proofs in appendix.
3. Authors have done comprehensive evaluation and ablation across different model families (Phi-3, Llama-2/3).

Weaknesses:
1. Technical novelty: While the combined optimization framework is novel, the individual components (RPCA for compression, policy gradient methods) are well-established.
2. Scalability concerns: Despite improvements, the bilevel optimization requires multiple sampling iterations and may face challenges scaling to very large models (>100B parameters).
3. Limited Accuracy and CPU SpeedUp compared to baseline: Zero shot accuracy gain not much significant and also inference speedup looks minimal compared to other methods.
4. Hyperparameter sensitivity: The method introduces several hyperparameters which might be very difficult to tune. Also ablation for some of the important hp (for example for all the hp of truncated Gaussian distribution) gamma should be provided.
5. Comparisons with some recent structured/unstructured pruning like GBLM-Pruner[1], ATP[2].
6. Inference speedups on GPUs are not shown.
7. Missing Convergence Analysis for the overall bilevel optimization.

References:

1. Das, Rocktim Jyoti, et al. "Beyond size: How gradients shape pruning decisions in large language models." arXiv preprint arXiv:2311.04902 (2023).

2. Huang, Weizhong, et al. "Determining Layer-wise Sparsity for Large Language Models Through a Theoretical Perspective." arXiv preprint arXiv:2502.14770 (2025).

---

> ### Author Rebuttal · Authors · 2025-07-31
>
> ## W1: Novel framework and well-established components.
> We thank the reviewer for acknowledging the novelty of our framework.
>
> While the individual components are well-established, our contribution lies in designing a collaborative mechanism and solving strategy that coordinate them under a bilevel framework, which, to the best of our knowledge, is the first of its kind in model pruning. We believe such collaboration framework would be effective and helpful to developing high-precision pruning methods for LLMs, especially in the scenario with limited computational and data resources.
>
> Moreover, beyond the general framework, we incorporate several specific design choices to enhance training efficiency. For instance, we adopt a truncated Gaussian prior, which effectively mitigates the risk of gradient explosion (see Section 4.2). This issue can arise when using a standard Gaussian distribution, as its gradient contains a $1/\sigma^2$ term. To ensure convergence, one typically needs to reduce $\sigma$ to concentrate the sampling distribution, but this can lead to unstable gradients. The truncated version addresses this problem by allowing us to concentrate samples via truncation while keeping $\sigma$ fixed, thereby maintaining gradient stability.
>
> ## W2: Scalability concerns.
> Thank you for pointing out this concern. Due to the space limit, please refer to our response to reviewer ZdES's W2 & Q1.
>
> ## W3: Limited improvement on zero shot and speedup.
>
> Regarding zero-shot accuracy, we would like to clarify two key points.
>  - Zero-shot task is relatively easy. At low sparsity levels (e.g., below 50%), the performance degradation caused by pruning is generally minor—even for baseline methods, the post-pruning performance remains close to that of the original model. This suggests that the room for further improvement is quite limited, making it nearly impossible to achieve significant performance gains under such settings.
>
> - Our method demonstrates clear and significant improvements under high sparsity levels (60–70%), as shown in Table 2 of the main paper and **Table R1** below.
>
> #### Table R1. Performance comparison under high prune rates.
> |Prune rate | Method | phi-3 wiki | phi-3 MMLU |phi-3 zero-shot | llama2-7b wiki | llama2-7b MMLU | llama2-7b zero-shot | llama3-8b wiki | llama3-8b MMLU | llama3-8b zero-shot |
> |---|---|---|---|---|---|---|---|---|---| ---|
> | 60% |SparseGPT | 51.44 | 37.37 | 54.61 | 18.38 | 29.77 | 57.92 | 23.21 | 34.23 | 55.39 |
> | 60% | Wanda | 52.28 | 36.96 | 55.48 | 18.12 | 30.41 | 57.64 | 23.63 | 31.82 | 54.86 |
> | 60% | OATS | 42.37 | 43.96 | 57.82 |16.78  | 34.64 | 59.43 | 21.30 | 37.26 | 57.78 |
> | 60% | QR |48.8  | 38.22 | 56.01 | 16.92 | 32.99 | 58.61 | 20.03 | 34.06 | 56.62 |
> | 60% | Ours | 32.46 | 52.27 | 59.24 | 15.07 | 39.78 | 60.52 | 17.03 | 43.91 | 59.34 |
> | 70% |SparseGPT | 1175 | 25.83 | 42.34 | 112.28 | 22.88 | 43.37 | 98.28 | 22.88 | 41.37 |
> | 70% | Wanda | 2124.9 | 26.01 | 38.94 | 235.7 | 25.3 | 39.71 | 183.0 | 25.45 | 38.15 |
> | 70% | OATS | 762.4 | 26.73 | 45.69 | 88.56 | 25.51 | 46.31 | 89.27 | 26.85 | 43.28 |
> | 70% | QR | 1375 | 25.5 | 40.88 | 122.9 | 24.53 | 42.49 | 111.37 | 26.8 | 41.14 |
> | 70% | Ours | 208.7 | 30.27 | 49.53 | 47.21 | 29.38 | 51.62 | 58.34 | 27.6 | 47.04 |
>
> Regarding inference speedup, we argue that
> - We present the results in Table 4 to illustrate the additional speedup gained from the low rank and sparse structure; specifically, low rank matrices are generally more hardware-friendly for acceleration, rather than highlighting their overall superiority in speedup.
> - At the same compression rates, the compressed models obtained from different pruning strategies usually yield comparable inference speedups, as the additional speedup derived from the unique matrix structure cannot be excessively high.
> - Consequently, our objective is to achieve higher accuracy than the baselines while maintaining the same compression ratio, as demonstrated by the results presented in the main text.
>
> ## W4 & Q3: Hyperparameter sensitivity.
> We acknowledge that our bilevel framework introduces several hyperparameters. However, we would like to clarify following facts to show that our algorithm is not sensitive to these hyperparameters.
> - In fact, we tuned these hyperparameters on Phi-3 mini and applied the same configuration across all other model experiments. Besides, parameters such as the initialization of the truncated Gaussian mean $\mu$ and the number of inner-loop iterations are consistent with those used in OATS.
> - We also conduct ablation studies on these key hps, and the results are presented in **Table R2** below.
> - Moreover, theoretical analysis (see our response to reviewer tY2u's Question 2) shows the dependency of convergence on the parameters such as $\gamma$.
> #### Table R2. Ablation study of hyperparameters Phi3-mini with 50% prune rate, wikitext2 PPL
> |$\sigma^2$ | 1e-3 | 5e-3 | 1e-2 | 5e-2 | 1e-1 |
> |---|---|---|---|---|---|
> | PPL | 15.16 | 14.94 | 14.87 | 14.92 | 15.07 |
> |$\gamma_0$ | **1e-3** | **5e-3** | **1e-2** | **5e-2** | **1e-1** |
> | PPL | 15.22 | 15.01 | 14.93 | 14.87 | 14.99 |
> |$\gamma_{target}$ | **1e-3** | **5e-3** | **1e-2** | **5e-2** | **1e-1** |
> | PPL | 14.88 | 14.87 | 14.90 | 15.07 | 15.18 |
>
> ## W5: Comparisons with GBLMpruner & ATP.
> Thank you for pointing out these recent works in model compression. We present the comparison results in **Table R3**.
>
> It is worth clarifying that ATP originally uses a customed and simplified version of the `lm_eval` package (based on version 0.3.0). Since we use an updated version (0.4.2) consistent with OATS in our main experiments, there may be slight discrepancies in evaluation results, which is also mentioned in OATS's Appendix A.12. To ensure fairness, we re-evaluated the perplexity on Wikitext2 of our methods using the version adopted in ATP's implementation.
>
> Due to our learned sparsity allocation and the integration of low-rank matrices, our method outperforms both GBLM-Pruner and ATP across all settings.
> #### Table R3. Comparison with GBLM and ATP on LLaMA2-7B. "-" means missing results in the original paper.
> |Prune rate | Method | PPL$\downarrow$ | MMLU$\uparrow$ | Zero-shot$\uparrow$ |
> |---|---|---|---|---|
> | 0%  | Dense | 5.47 | 50.12 | 66.27 |
> | 50% | GBLMpruner | 6.86 | - | - |
> | 50% | ATP+wanda | 6.82 | - | 64.49 |
> | 50% | Ours | 6.47 | 46.10 | 65.34 |
> | 60% | ATP+wanda | 9.15 | 32.80 |58.63|
> | 60% | Ours | 8.79 | 39.78 | 60.52 |
>
> ## W6 & Q1: GPU Inference speedups.
> We appreciate the reviewer’s comment and agree that presenting GPU inference speedup is important. In our initial experiments, we followed the evaluation protocols of OATS and OWL, which focus on CPU speed up, and therefore we did not include GPU results.
> To address this, we followed the ATP protocol and conducted GPU inference tests using `nm-vllm`. The results are presented in **Table R4**.
> Compared with unstructured methods, our approach, Low rank & Sparse, leads to more practically meaningful GPU inference acceleration.
> #### Table R4 GPU inference speedup LLaMA2-13B, on a single 80 GB A100 GPU
> |Prune rate|Prune Type|Method|Throughput (tokens/s)|Speedup|Perplexity |
> |---|---|---|---|---|---|
> | 0% | Dense | - | 47.31 | 1.0 | 7.91 |
> | 40% | Unstructured | Wanda | 68.73 | 1.45 | 8.77 |
> | 40% | Low rank & Sparse | Ours | 79.87 | 1.69 | 8.31 |
> | 50% | Unstructured | Wanda | 83.15 | 1.76 | 10.05 |
> | 50% | Semi-structured | Wanda | 92.36 | 1.95 | 16.53 |
> | 50% | Low rank & Sparse | Ours | 102.45 | 2.17 | 9.23 |
>
> ## W7 & Q2: Convergence Analysis.
> Due to space limitations in the responses, we provide a detailed convergence analysis of the bilevel framework in our response to **Reviewer tY2u**'s Question 2. We will include this analysis in the appendix of the revised version if accepted.
>
> Theoretical analysis demonstrate that a smaller value of $\gamma$ leads to reduced gradient variance, which in turn facilitates faster convergence of the optimization process.
>
>
> ## Q4: Time taken compared to one-shot methods:
> We acknowledge that, unlike one‑shot methods such as Wanda or SparseGPT, our bilevel framework incurs additional runtime. However, our entire process still completes within a few hours on standard hardware, which is always affordable in downstearm applications.
>
> Importantly, model compression is a one‑time offline cost, where **peak GPU memory usage often outweighs total runtime in practical deployments. Our method uses no more memory than one‑shot approaches but delivers superior accuracy and inference speedups**, making the extra time investment worthwhile.
> The results of compression time are summarized in **Table R5**. Even **QR**, a simplified version of our method, still outperforms SparseGPT and Wanda under comparable time cost.
>
> #### Table R5. Compression time, Phi-3 mini 60% prune rate
> |Method | Prune Time | Wiki | MMLU | Zero-shot |
> |---|---|---|---|---|
> |SparseGPT | 27min | 51.44 | 37.37 | 54.61 |
> |Wanda | 12min | 52.28 | 36.96 | 55.48 |
> |OATS | 5h | 42.37 | 43.96 | 57.82 |
> |QR | 15min |48.8  | 38.22 | 56.01 |
> |Ours | 4.5h | 32.46 | 52.27 | 59.24 |

---

> > ### Comment · Reviewer_AQTX · 2025-08-07
> > **Response**
> >
> > Thank you for your detailed rebuttal, which addresses several of my key concerns with substantial additional experiments and theoretical analysis. The GPU inference benchmarking on A100 demonstrates meaningful acceleration, the formal convergence analysis fills an important theoretical gap, and the high-sparsity performance results  are impressive. However, my major concerns remain regarding the practical burden of hyperparameter tuning compared to one-shot methods, and the significant computational overhead (4.5h vs 12-27min) that may limit deployment scenarios. I will retain my positive score.

---

### Note · Authors · 2025-08-13

Dear ACs and Reviewers,

Thanks for your time and valuable comments. We would like to provide our final remarks below.
- The initial score of our paper was 54443. After rebuttal:
    - Reviewer sRwV(5) **does not have any more questions** with our **thoughtful clarifications and additional results**.
    - Reviewer tY2u(4) became **more convinced about the theoretical justification**.
    - Reviewers AQTX and ZdES(4) indicated that we had resolved their major concerns while maintaining a positive evaluation.
    - Rreviewer nSyU(3) also expressed that he had **no further questions**.
- The main contributions of our paper have been widely recognized by the reviewers:
  - We propose a novel optimization framework that enables effective collaboration between learning and matrix approximation for LLM compression, acknowledged by reviewers AQTX, tY2u, ZdES, and sRwV.
  - Our optimization techniques, including policy gradient and truncated Gaussian prior, are considered well-designed and novel; reviewer ZdES noted they “could advance the state-of-the-art in LLM compression.”
  - Our method is supported by a theoretical foundation, with our detailed analysis receiving broad recognition from reviewers AQTX, tY2u, ZdES, and sRwV.
  - Reviewers AQTX, ZdES, and sRwV agreed with our comprehensive evaluations across multiple models and our ablation studies validating the effectiveness of our algorithm.
- We had a thorough discussion with reviewer nSyU (initial score being 3) covering the following points:
    - **GPU speedup.** In the rebuttal, we provided detailed GPU acceleration results. Moreover, sparse + low-rank structures are widely recognized in the community as an effective model compression paradigm to achieve practical speedup on different platforms [1,OATS]. Building on this foundation, we propose more advanced pruning methods within this paradigm, with further acceleration expected as matrix computation libraries improve in the future.
    - **Diagonal approximation.** Our main contribution lies in the effective collaboration between learning and matrix approximation. Diagonal approximation for $X^\top X$, used in prior works such as OATS and Wanda to capture outlier features, was adopted for fair comparison. Results using alternative method were given in rebuttal.

Once again, we thank the ACs and all reviewers for your thoughtful feedback and hard work.

[1]Zhuo Li et al. "Model compression for deep neural networks: A survey."

---

### Decision · Program_Chairs · 2025-09-17

**Decision:**

Accept (poster)

**Comment:**

This paper presents a novel and well-motivated bilevel optimization framework for Large Language Model (LLM) compression, integrating learning-based sparsity allocation with matrix approximation. The work makes significant technical contributions, and its initial strengths were substantially reinforced by the author's response during the discussion period that addressed most of reviewers concerns with new experiments and analysis.

There are two key reasons for acceptance: (1) Originality and technical innovation: Reviewers consistently mentioned the novelty of the bilevel framework and its core technical contributions. This includes a truncated Gaussian prior-based probabilistic parameterization with a policy gradient estimator for the outer loop, and an efficient QR-based RPCA algorithm for inner loop computations. The theoretical foundation, including convergence analysis, has been well-addressed. (2) Strong empirical performance: The method demonstrates superior performance in perplexity, MMLU, and zero-shot accuracy across various LLMs (Phi-3, Llama-2/3), particularly excelling under high sparsity levels.

There is still one important concern regarding the actual speed-up achievable with a GPU. AC believes this concern remains valid; however, it does not diminish the paper's empirical success in outperforming other unstructured and semi-structured pruning methods. AC requests that the authors accurately and thoroughly document the limitations of speed-up when using GPUs, ensuring that the community is fully informed about potential directions for future improvements.